# TRANSFERRING LEARNING TRAJECTORIES OF NEURAL NETWORKS

**Daiki Chijiwa**[*]
NTT Computer and Data Science Laboratories, NTT Corporation

## ABSTRACT

Training deep neural networks (DNNs) is computationally expensive, which is problematic especially when performing duplicated or similar training runs in model ensemble or fine-tuning pre-trained models, for example. Once we have trained one DNN on some dataset, we have its learning trajectory (i.e., a sequence of intermediate parameters during training) which may potentially contain useful information for learning the dataset. However, there has been no attempt to utilize such information of a given learning trajectory for another training. In this paper, we formulate the problem of "transferring" a given learning trajectory from one initial parameter to another one (named *learning transfer problem*) and derive the first algorithm to approximately solve it by matching gradients successively along the trajectory via permutation symmetry. We empirically show that the transferred parameters achieve non-trivial accuracy before any direct training, and can be trained significantly faster than training from scratch.

## 1 INTRODUCTION

Enormous computational cost is a major issue in deep learning, especially in training large-scale neural networks (NNs). Their highly non-convex objective and high-dimensional parameters make their training difficult and inefficient. Toward a better understanding of training processes of NNs, their loss landscapes (Hochreiter & Schmidhuber, 1997; Choromanska et al., 2015) have been actively studied from viewpoints of optimization (Haeffele & Vidal, 2017; Li & Yuan, 2017; Yun et al., 2018) and geometry (Freeman & Bruna, 2017; Simsek et al., 2021). One of the geometric approaches to loss landscapes is mode connectivity (Garipov et al., 2018; Draxler et al., 2018), which shows the existence of low-loss curves between any two optimal solutions trained with different random initializations or data ordering. This indicates a surprising connection between seemingly different independent trainings.

Linear mode connectivity (LMC), a special case of mode connectivity, focuses on whether or not two optimal solutions are connected by a low-loss linear path, which is originally studied in relation to neural network pruning (Frankle et al., 2020). It is known that the solutions trained from the same initialization (and data ordering in the early phase) tend to be linearly mode connected (Nagarajan & Kolter, 2019; Frankle et al., 2020), but otherwise they cannot be linearly connected in general. However, Entezari et al. (2021) observed that even two solutions trained from different random initializations can be linearly connected by an appropriate permutation symmetry. Ainsworth et al. (2023) developed an efficient method to find such permutations and confirmed the same phenomena with modern NN architectures. These observations strengthen the expectation on some sort of similarity between two independent training runs even from different initializations, via permutation symmetry.

In this paper, motivated by these observations, we make the first attempt to leverage such similarity between independent training processes for efficient training. In particular, we introduce a novel problem called *learning transfer problem*, which aims to reduce training costs for seemingly duplicated training runs on the same dataset, such as model ensemble or knowledge distillation, by transferring a learning trajectory for one initial parameter to another one without actual training. The problem statement is informally stated as follows:

---

[*]Corresponding author: `daiki.chijiwa@ntt.com`

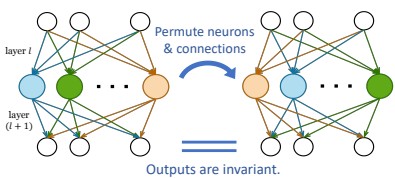

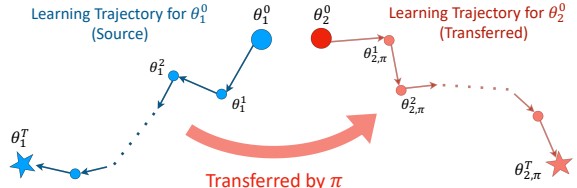

Figure 1: Permutation symmetry of neural networks. (Section 2.2)

Figure 2: Transfer a learning trajectory from one to another NN by a permutation symmetry $\pi$. (Section 3)

**Learning transfer problem** (informal). *Suppose that a source learning trajectory $(\theta_1^0, \cdots, \theta_1^T)$ is given for some initial parameter $\theta_1^0$. Given another initial parameter $\theta_2^0$, called target initialization, how can we synthesize the learning trajectory $(\theta_2^0, \cdots, \theta_2^T)$ for $\theta_2^0$ efficiently?*

To tackle this problem, as illustrated in Figure 2, we take an approach to transform the source trajectory $(\theta_1^0, \cdots, \theta_1^T)$ by an appropriate permutation symmetry $\pi$ as in the previous works of LMC. In Section 3, we formulate the learning transfer problem as a non-linear optimization problem for $\pi$. We also investigate how much the source trajectory for $\theta_1^0$ can be transformed close to the target trajectory for $\theta_2^0$ by the optimal $\pi$, both theoretically and empirically. We then derive a theoretically-grounded algorithm to approximately solve it, and also develop practical techniques to reduce its storage and computational cost. Since our final algorithm requires only several tens of gradient computations and lightweight linear optimization in total, we can transfer a given source trajectory very efficiently compared to training from scratch. In Section 4, first we empirically demonstrate that learning trajectories can be successfully transferred between two random or pre-trained initializations, which leads to non-trivial accuracy without direct training (Section 4.1). Next we further confirmed that the transferred parameters can indeed accelerate the convergence in their subsequent training (Section 4.2). Finally, we investigate what properties the transferred parameters inherit from the target initializations (Section 4.3). Surprisingly, we observed that, even when the source trajectory is trained from poorly generalizing initialization $\theta_1^0$ (and thus inherits poor generalization), the one transferred onto more generalizing initialization $\theta_2^0$ acquires better generalization inherited from $\theta_2^0$.

In summary, our contributions are as follows: (1) we formulated the brand new problem of learning transfer with theoretical evidence, (2) derived the first algorithm to solve it, (3) empirically showed that transferred parameters can achieve non-trivial accuracy without direct training and accelerate their subsequent training, and (4) investigated the benefit/inheritance from target initializations. Finally, more related works are discussed in Appendix C.

## 2 BACKGROUND

### 2.1 NEURAL NETWORKS

Let $L, N \in \mathbb{N}$. Let $f(x; \theta)$ be an $L$-layered neural network (NN) parameterized by $\theta \in \mathbb{R}^N$ with a non-linear activation function $\sigma : \mathbb{R} \to \mathbb{R}$ and intermediate dimensions $(d_0, \cdots, d_L) \in \mathbb{N}^{L+1}$. Given an input $x \in \mathbb{R}^{d_0}$, the output $f(x; \theta) := x_L \in \mathbb{R}^{d_L}$ is computed inductively as follows:

$$x_i := \begin{cases} x, & (i = 0) \\ \sigma(W_i x_{i-1} + b_i), & (1 \le i \le L-1) \\ W_L x_{L-1} + b_{L-1}, & (i = L) \end{cases}$$

where $W_i \in \mathbb{R}^{d_i \times d_{i-1}}, b_i \in \mathbb{R}^{d_i}$ are weight matrices and bias vectors. Under these notation, the parameter vector $\theta$ is described as $\theta = (W_1, b_1, \cdots, W_L, b_L) \in \mathbb{R}^N$.

Stochastic gradient descent (SGD) is a widely used approach for training neural networks. Let $\mathcal{X}$ be the input space $\mathbb{R}^{d_0}$ and $\mathcal{Y}$ be the output space $\mathbb{R}^{d_L}$. Let $\mathcal{D}$ be a probabilistic distribution over the input-output space $\mathcal{X} \times \mathcal{Y}$, and $\mathcal{L} : \mathcal{Y} \times \mathcal{Y} \to \mathbb{R}$ be a differentiable loss function. SGD trains a neural network $f(x; \theta)$ by iterating the following steps: (1) Sampling a mini-batch $B = ((x_1, y_1), \cdots, (x_b, y_b)) \sim \mathcal{D}^b$ of size $b \in \mathbb{N}$, (2) computing an estimated gradient $g_B := \frac{1}{b} \sum_{i=1}^b \nabla_\theta \mathcal{L}(f(x_i; \theta), y_i)$ for the mini-batch and (3) updating the model parameter $\theta$ by $\theta - \alpha g_B + (\text{momentum})$ where $\alpha \in \mathbb{R}_{>0}$ is a fixed or scheduled step size.

## 2.2 PERMUTATION SYMMETRY OF NNS

For simplicity, we assume that all bias vectors $b_i$ are zero by viewing them as a part of the weight matrices. Let $\Theta$ be the parameter space $\{\theta = (W_1, \cdots, W_L) \in \mathbb{R}^N\}$ for the $L$-layered neural network $f(x; \theta)$. Now we introduce a permutation group action on the parameter space $\Theta$. For $n \in \mathbb{N}$, let $S_n$ denotes the symmetric group over $\{1, \cdots, n\} \subset \mathbb{N}$, which is the set of all bijective mapping $\sigma : \{1, \cdots, n\} \to \{1, \cdots, n\}$. We define our permutation group $G$ by $G := S_{d_1} \times \cdots \times S_{d_{L-1}}$. The group action $G \times \Theta \to \Theta$ is defined as follows: For $\pi = (\sigma_1, \cdots, \sigma_{L-1}) \in G$ and $\theta \in \Theta$, the action $\pi\theta$ is defined by

$$\pi\theta := (\sigma_1 W_1, \cdots, \sigma_i W_i \sigma_{i-1}^{-1}, \cdots, W_L \sigma_{L-1}^{-1}) \in \Theta, \tag{1}$$

where each $\sigma_i$ is viewed as the corresponding permutation matrix of size $d_i \times d_i$. We call this group action the permutation symmetry of $L$-layered neural networks.

Simply speaking, the action $\pi\theta$ just interchanges the axes of the intermediate vector $x_i$ of the neural network $f(x; \theta)$ with the corresponding base change of the weight matrices and bias vectors (Figure 1). Thus we can see that this action does not change the output of the neural network, i.e., $f(x; \pi\theta) = f(x; \theta)$ for every $x \in \mathcal{X}, \theta \in \Theta, \pi \in G$. In other words, the two parameters $\theta$ and $\pi\theta$ can be identified from the functional perspective of neural networks.

## 2.3 PARAMETER ALIGNMENT BY PERMUTATION SYMMETRY

Previous work by Ainsworth et al. (2023) attempts to merge given two NN models into one model by leveraging their permutation symmetry. They reduced the merge problem into the parameter alignment problem:

$$\min_{\pi \in G} \|\pi\theta_1 - \theta_2\|_2^2 = \min_{\pi = (\sigma_1, \cdots, \sigma_{L-1})} \sum_{1 \le i \le L} \|\sigma_i W_i \sigma_{i-1}^{-1} - Z_i\|_F^2, \tag{2}$$

where $\theta_1 = (W_1, \cdots, W_L)$ and $\theta_2 = (Z_1, \cdots, Z_L)$ are the parameters to be merged. To solve this, they also proposed a coordinate descent algorithm by iteratively solving the following linear optimizations regarding to each $\sigma_i$'s:

$$\max_{\sigma_i \in S_i} \langle \sigma_i, Z_i \sigma_{i-1} W_i^\top + Z_{i+1}^\top \sigma_{i+1} W_{i+1} \rangle \tag{3}$$

The form of this problem has been well-studied as a linear assignment problem, and we can solve it in a very efficient way (Kuhn, 1955). Although the coordinate descent algorithm was originally proposed for model merging, we can also use it for other problems involving the parameter alignment problem (eq. 2).

# 3 LEARNING TRANSFER

In this section, first we formulate the problem of transferring learning trajectories (which we call *learning transfer problem*) as a non-linear optimization problem. Next, we derive an algorithm to solve it by reducing the non-linear optimization problem to a sequence of linear optimization problems. Finally, we introduce additional techniques for reducing the storage and computation cost of the derived algorithm.

## 3.1 PROBLEM FORMULATION

Let $f(x; \theta)$ be some NN model with an $N$-dimensional parameter $\theta \in \mathbb{R}^N$. A sequence of $N$-dimensional parameters $(\theta^0, \cdots, \theta^T) \in \mathbb{R}^{N \times (T+1)}$ is called a *learning trajectory* of length $T$ for the neural network $f(x; \theta)$ if the training loss for $\theta^t$ decreases as $t$ increases, from the initial parameter $\theta^0$ to the convergent one $\theta^T$. Note that *we do not specify what $t$ represents a priori; it could be iteration, epoch or any notion of training timesteps.*[1] Now we can state our main problem:

---

[1] In our experiments, due to practical constraints, we mainly consider cases where $t$ represents training epoch or linear interpolating step as explained in Section 3.4, which leads to the length $T$ being smaller than a hundred.

**Learning transfer problem** (informal). Suppose that a learning trajectory $(\theta_1^0, \cdots, \theta_1^T)$ is given for an initial parameter $\theta_1^0$, which we call a *source trajectory*. Given another initial parameter $\theta_2^0$ "similar" to $\theta_1^0$ in some sense, how can we synthesize the learning trajectory $(\theta_2^0, \cdots, \theta_2^T)$, which we call a *transferred trajectory*, for the given initialization $\theta_2^0$? (Figure 2)

To convert this informal problem into a computable one, we need to define the notion of "similarity" between two initial parameters. As a first step, in this paper, we consider two initializations are "similar" if two learning trajectories starting from them are indistinguishable up to permutation symmetry of neural networks (Section 2.2). In other words, for the two learning trajectories $(\theta_1^0, \cdots, \theta_1^T)$ and $(\theta_2^0, \cdots, \theta_2^T)$, we consider the following assumption:

**Assumption (P).** There exists a permutation $\pi$ satisfying $\pi(\theta_1^t - \theta_1^{t-1}) \approx \theta_2^t - \theta_2^{t-1}$ for $t = 1, \cdots, T$, where the transformation $\pi(\theta_1^t - \theta_1^{t-1})$ is as defined in Section 2.2.

Under this assumption, if we know the permutation $\pi$ providing the equivalence between the source and transferred trajectories, we can recover the latter one $(\theta_2^0, \cdots, \theta_2^T)$ from the former one $(\theta_1^0, \cdots, \theta_1^T)$ and the permutation $\pi$, by setting $\theta_2^t := \theta_2^{t-1} + \pi(\theta_1^t - \theta_1^{t-1})$ inductively on $t$ (Figure 2). Therefore, the learning-trajectory problem can be reduced to estimating the permutation $\pi$ from the source trajectory $(\theta_1^0, \cdots, \theta_1^T)$ and the given initialization $\theta_2^0$.

Naively, to estimate the permutation $\pi$, we want to consider the following optimization problem:

$$\min_\pi \sum_{t=1}^T \left\| \pi(\theta_1^t - \theta_1^{t-1}) - (\theta_2^t - \theta_2^{t-1}) \right\|_2^2 \tag{4}$$

However, this problem is ill-defined in our setting since each $\theta_2^t$ is not available for $1 \leq t \leq T$ in advance. Even if we defined $\theta_2^t := \theta_2^{t-1} + \pi(\theta_1^t - \theta_1^{t-1})$ in the equation (4) as discussed above, the optimization problem became trivial since any permutation $\pi$ makes the $L^2$ norm to be zero.

Thus we need to fix the optimization problem (eq. 4) not to involve unavailable terms. We notice that the difference $\theta_2^t - \theta_2^{t-1}$ can be roughly approximated by a negative gradient at $\theta_2^{t-1}$ averaged over a mini-batch if the trajectory is enough fine-grained. Therefore, we can consider the approximated version of equation (4) as follows:

$$\mathcal{P}_T: \quad \min_\pi \sum_{t=0}^{T-1} \left\| \pi \nabla_{\theta_1^t} \mathcal{L} - \nabla_{\theta_{2,\pi}^t} \mathcal{L} \right\|_2^2, \text{ where } \theta_{2,\pi}^t := \theta_{2,\pi}^{t-1} + \pi(\theta_1^t - \theta_1^{t-1}). \tag{5}$$

In contrast to the equation (4), this optimization problem is well-defined even in our setting because each $\theta_{2,\pi}^t$ is defined by using the previous parameter $\theta_{2,\pi}^{t-1}$ inductively.

### 3.2 WHEN AND HOW DOES ASSUMPTION (P) HOLDS?

Before delving into the optimization problem $\mathcal{P}_T$, here we briefly investigate when and how our Assumption (P) holds. To measure the similarity of two vectors $\pi(\theta_1^t - \theta_1^{t-1})$ and $\theta_2^t - \theta_2^{t-1}$, we mainly use a variant of normalized distance $\|v_1 - v_2\| / \sqrt{\|v_1\| \|v_2\|}$ for two vectors $v_1, v_2 \in \mathbb{R}^n$, which can also be considered as cosine distance when $\|v_1\| \approx \|v_2\|$.

First of all, we study the case of 2-layered ReLU neural networks $f_{w,v}(x) := \sum_{i=1}^N v_i \sigma(\sum_{j=1}^d w_{ij} x_j)$ with $N$ hidden neurons, where $\sigma(z) := \max(z, 0)$ is ReLU activation. Let $\theta_1^0 = ((w_{ij})_{ij}, (v_i)_i)$ and $\theta_2^0 = ((w'_{ij})_{ij}, (v'_i)_i)$ be initialized by Kaiming uniform initialization (He et al., 2015), i.e., $w_{ij}, w'_{ij} \sim U([-\frac{1}{\sqrt{d}}, \frac{1}{\sqrt{d}}])$ and $v_i, v'_i \sim U([-\frac{1}{\sqrt{N}}, \frac{1}{\sqrt{N}}])$. Then it is theoretically shown that Assumption (P) holds at initialization with high probability when the hidden dimension $N$ is sufficiently large:

**Theorem 3.1 (See Theorem A.1 for details)** *Given two pairs of randomly initialized parameters $(w, v)$ and $(w', v')$, with high probability, there exists a permutation symmetry $\pi \in S_N$ such that the normalized distance between the expected gradients $\mathbb{E}_{(x,y)}[\nabla_{w,v}\mathcal{L}]$ and $\mathbb{E}_{(x,y)}[\nabla_{w'',v''}\mathcal{L}]$, where $(w'', v'')$ is the permuted one of $(w', v')$ by $\pi$, can be arbitrarily small when $N$ is sufficiently large.*

Although this result only validates Assumption (P) at initialization, we can naturally expect that same similarity holds for more iterations ($t > 0$) due to (almost everywhere) smoothness of neural networks with respect to parameters. Figure (3a) empirically validates this expectation.

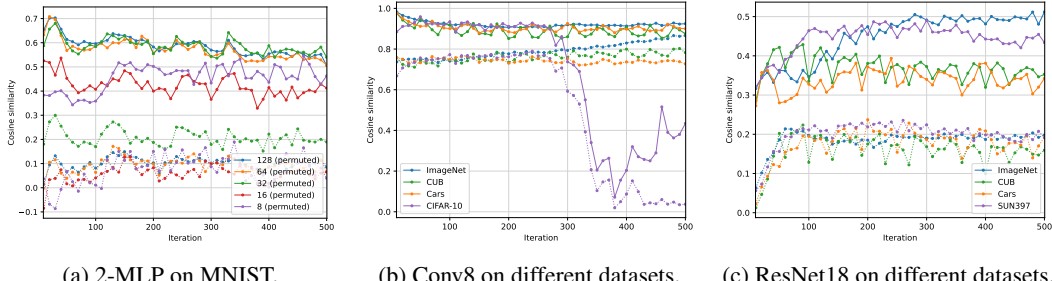

(a) 2-MLP on MNIST.  (b) Conv8 on different datasets.  (c) ResNet18 on different datasets.

Figure 3: We evaluate cosine similarities between $\pi(\theta_1^t - \theta_1^{t-1})$ and $\theta_2^t - \theta_2^{t-1}$, where $\theta_1^t$ or $\theta_2^t$ represents a parameter trained for $10t$ iterations, averaged over timesteps $t$. Solid lines are plotted for the solution $\pi$ of equation (4), and dotted lines are for $\pi$ being the identity transformation (i.e., the case not permuted) as a baseline. Fig.(a): Experiments using 2-layered MLP with various hidden dimensions. Over-parameterization leads to higher cosine similarities of learning trajectories. Fig.(b, c): Experiments with modern network architectures (8-layered CNN / ResNet18) on different datasets.

Also, we empirically investigate the case of modern network architectures in Figure (3b), (3c). The results show that the overall trends of similarity between learning trajectories heavily depend on network architecture rather than datasets, which is also indicated by Theorem 3.1. Moreover, additional experiments in Appendix E.8 show that Assumption (P) also holds even when the initialization is pre-trained, instead of a random one, in the very early phase of training. Nevertheless, it remains for future work to make the assumption more appropriate for modern architectures so that it holds for longer iterations beyond initialization, for example in the case of Conv8 on CIFAR-10 in Figure (3b), or in the case of pre-trained initializations rather than random initializations.

### 3.3 ALGORITHM: GRADIENT MATCHING ALONG TRAJECTORY

Now our goal is to solve the optimization problem $\mathcal{P}_T$ (eq. 5). However, the problem $\mathcal{P}_T$ seems hard to solve directly because the variable $\pi$ appears non-linearly in the second term $\nabla_{\theta_{2,\pi}^t} \mathcal{L}$. To avoid the non-linearity, we introduce a sequence of linear sub-problems $\{\mathcal{P}'_s\}_{1 \le s \le T}$ whose solution converges to the solution for $\mathcal{P}_T$. For each $s \in \{1, \cdots, T\}$, we consider the following problem:

$$\mathcal{P}'_s : \quad \min_{\pi_s} \sum_{t=0}^{s-1} \left\| \pi_s \nabla_{\theta_1^t} \mathcal{L} - \nabla_{\theta_{2,\pi_{s-1}}^t} \mathcal{L} \right\|_2^2 \tag{6}$$

Since the second term in $\mathcal{P}'_s$ uses the solution $\pi_{s-1}$ for the previous sub-problem $\mathcal{P}'_{s-1}$, the unknown variable $\pi_s$ appears only in the first term $\pi_s \nabla_{\theta_1^t} \mathcal{L}$ in a linear way. Moreover, the following lemma implies that the final solution $\pi_T$ from the sequence $\{\mathcal{P}'_s\}_{1 \le s \le T}$ approximates the solution for the original problem $\mathcal{P}_T$:

**Lemma 3.2** *Under some regularity assumption, we have $\theta_{2,\pi_s}^t \approx \theta_{2,\pi_{s'}}^t$ for $0 \le t \le s < s'$.*

The proof will be given in Appendix B. Indeed, by this approximation, we find out that the solution $\pi_T$ for the last sub-problem $\mathcal{P}'_T$ minimizes

$$\sum_{t=0}^{T-1} \left\| \pi_T \nabla_{\theta_1^t} \mathcal{L} - \nabla_{\theta_{2,\pi_{T-1}}^t} \mathcal{L} \right\|_2^2 \approx \sum_{t=0}^{T-1} \left\| \pi_T \nabla_{\theta_1^t} \mathcal{L} - \nabla_{\theta_{2,\pi_T}^t} \mathcal{L} \right\|_2^2, \tag{7}$$

where the right-hand side is nothing but the objective of the original problem $\mathcal{P}_T$.

Algorithm 1 gives a step-by-step procedure to obtain the transferred learning trajectory $(\theta_2^1, \cdots, \theta_2^T)$ by solving the sub-problems $\{\mathcal{P}'_s\}_{0 \le s \le T}$ sequentially. In lines 2-6, it computes an average of gradients $\nabla_\theta \mathcal{L}$ over a single mini-batch for each $\theta = \theta_1^{t-1}, \theta_2^{t-1}, (1 \le t \le s)$, which is required in the $s$-th sub-problem $\mathcal{P}'_s$ (eq. 6). In line 7, the $s$-th permutation $\pi_s$ is obtained as a solution of the sub-problem $\mathcal{P}'_s$, which can be solved as a linear optimization (eq. 3) using the coordinate descent algorithm proposed in Ainsworth et al. (2023). Then we update the transferred parameter $\theta_2^t$ for $t = 1, \cdots, s$ in line 8.

**Algorithm 1** Gradient Matching along Trajectory (**GMT**)

**Require:** $(\theta_1^0, \cdots, \theta_1^T) \in \mathbb{R}^{n \times (T+1)}, \theta_2^0 \in \mathbb{R}^n$
1: **for** $s = 1, \cdots, T$ **do**
2:      **for** $t = 1, \cdots, s$ **do**
3:          Sample $(x_1, y_1), \cdots, (x_b, y_b) \sim \mathcal{D}$.
4:          $g_1^t \leftarrow \frac{1}{b} \sum_{i=1}^b \nabla_{\theta_1^{t-1}} \mathcal{L}(f(x_i; \theta_1^{t-1}), y_i)$
5:          $g_2^t \leftarrow \frac{1}{b} \sum_{i=1}^b \nabla_{\theta_2^{t-1}} \mathcal{L}(f(x_i; \theta_2^{t-1}), y_i)$
6:      **end for**
7:      $\pi_s \leftarrow \arg\min_\pi \sum_{t=1}^s \|g_2^t - \pi g_1^t\|_2^2$
8:      **for** $t = 1, \cdots, s$ **do**
9:          $\theta_2^t \leftarrow \theta_2^{t-1} + \pi_s(\theta_1^t - \theta_1^{t-1})$
10:      **end for**
11: **end for**
12: return $(\theta_2^1, \cdots, \theta_2^s)$

**Algorithm 2** Fast Gradient Matching along Trajectory (**FGMT**)

**Require:** $(\theta_1^0, \cdots, \theta_1^T) \in \mathbb{R}^{n \times (T+1)}, \theta_2^0 \in \mathbb{R}^n$
1: **for** $s = 1, \cdots, T$ **do**
2:      Sample $(x_1, y_1), \cdots, (x_b, y_b) \sim \mathcal{D}$.
3:      $g_1^t \leftarrow \frac{1}{b} \sum_{i=1}^b \nabla_{\theta_1^{s-1}} \mathcal{L}(f(x_i; \theta_1^{s-1}), y_i)$
4:      $g_2^t \leftarrow \frac{1}{b} \sum_{i=1}^b \nabla_{\theta_2^{s-1}} \mathcal{L}(f(x_i; \theta_2^{s-1}), y_i)$
5:      $\pi_s \leftarrow \arg\min_\pi \sum_{t=1}^s \|g_2^t - \pi g_1^t\|_2^2$
6:      **for** $t = 1, \cdots, s$ **do**
7:          $\theta_2^t \leftarrow \theta_2^{t-1} + \pi_s(\theta_1^t - \theta_1^{t-1})$
8:      **end for**
9: **end for**
10: return $(\theta_2^1, \cdots, \theta_2^s)$

### 3.4 ADDITIONAL TECHNIQUES

While Algorithm 1 solves the learning transfer problem (eq. 5) approximately, it still has some issues in terms of storage and computation cost. Here we explain two practical techniques to resolve them.

**Linear trajectory.** In terms of the storage cost, Algorithm 1 requires a capacity of $T + 1$ times the model size to keep a learning trajectory of length $T$, which will be a more substantial issue as model size increases or the trajectory becomes fine-grained. To reduce the required storage capacity, instead of keeping the entire trajectory, we propose to imitate it by linearly interpolating the end points. In other words, given an initial parameter $\theta_1^0$ and the final $\theta_1^T$, we define a new trajectory $[\theta_1^0 : \theta_1^T] := (\theta_1^0, \cdots, \theta_1^t, \cdots, \theta_1^T)$ with $\theta_1^t := (1 - \lambda_t)\theta_1^0 + \lambda_t \theta_1^T$ and $0 = \lambda_0 \leq \cdots \leq \lambda_t \leq \cdots \leq \lambda_T = 1$.

Previous studies on the monotonic linear interpolation (Goodfellow et al., 2015; Frankle, 2020) indicate that such a linearly interpolated trajectory satisfies our definition of learning trajectories in SGD training of modern deep neural networks. Throughout this paper, we employ uniform scheduling for $\lambda_t$ defined by $\lambda_{t+1} - \lambda_t := 1/T$. Next, we compare the transferred results between the linear trajectory $[\theta_1^0 : \theta_1^T]$ and the actual trajectory $(\theta_1^0, \cdots, \theta_1^T)$ where each $\theta_1^t$ is a checkpoint at the $t$-th training epoch on CIFAR-10 with $T = 60$. Interestingly, the transfer of the linear trajectory is more stable and has less variance than the transfer of the actual one. This may be because the actual trajectory contains noisy information while the linear trajectory is directed towards the optimal solution $\theta_1^T$. Due to its storage

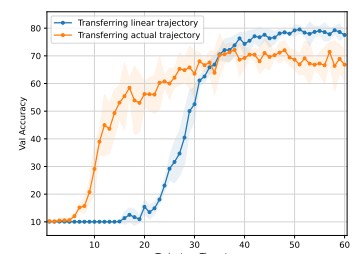

Figure 4: Linear vs. actual trajectory (Conv8 on CIFAR-10).

efficiency and stability in accuracy, we employ the linear trajectory of the length $T \leq 40$ throughout our experiments in Section 4.

**Gradient caching.** In terms of the computation cost, Algorithm 1 requires $O(T^2)$ times gradient computation. To reduce the number of gradient computation, we propose to cache the gradients once computed instead of re-computing them for every $s = 1, \cdots, T$. In fact, the cached gradients $\nabla_{\theta_{2,\pi_s}^{t-1}} \mathcal{L}$ and the re-computed gradients $\nabla_{\theta_{2,\pi_{s'}}^{t-1}} \mathcal{L}$ are not the same quantity exactly since the intermediate parameter $\theta_{2,\pi_s}^{t-1} = \theta_2^0 + \pi_s(\theta_1^{t-1} - \theta_1^0)$ takes different values for each $s$. However, they can be treated as approximately equal by Lemma 3.2 if we assume the continuity of the gradients. Now we can reduce the number of gradient computation from $O(T^2)$ to $O(T)$ by caching the gradients once computed. We describe this computationally efficient version in Algorithm 2.

## 4 EXPERIMENTS

In this section, we empirically evaluate how learning transfer works on standard vision datasets. First, we compare our proposed methods (**GMT**, **FGMT**) and two baselines (**Naive**, **Oracle**), which are

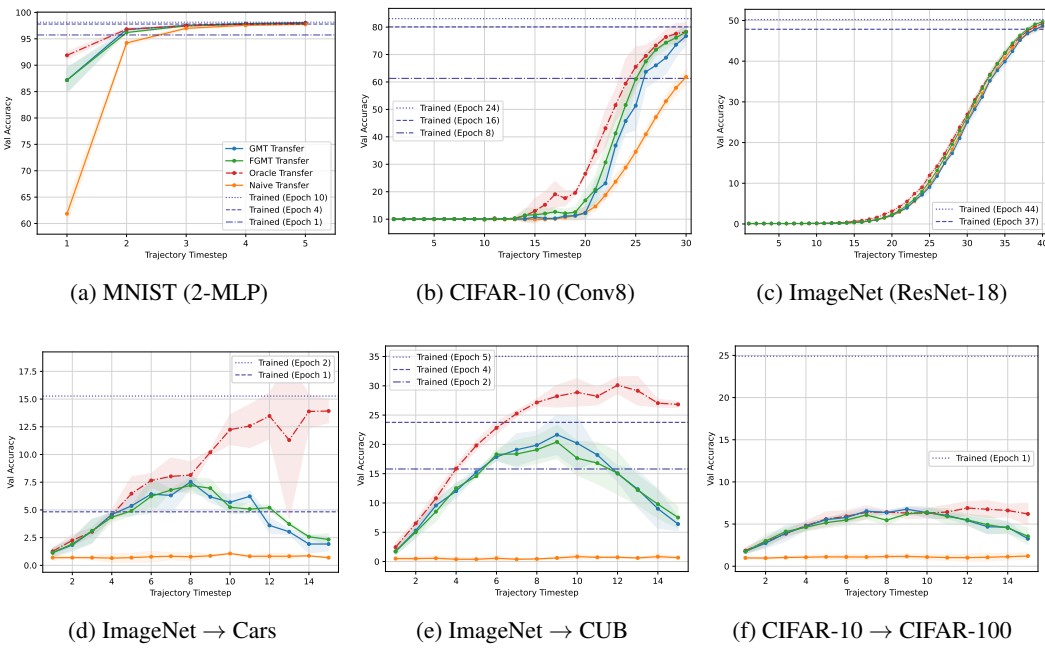

(a) MNIST (2-MLP)     (b) CIFAR-10 (Conv8)     (c) ImageNet (ResNet-18)

(d) ImageNet → Cars     (e) ImageNet → CUB     (f) CIFAR-10 → CIFAR-100

Figure 5: We plot the validation accuracies of the transferred parameter $\theta_{2,\pi_t}^t$ for each $t = 1, \cdots, T$ with various datasets and NN architectures. We also provide the standard deviation over three runs for each experiment. The dotted bars show accuracies in standard training. (Upper) Transfer of a learning trajectory on a single dataset between random initial parameters. (Lower) Transfer of a fine-tuning trajectory between pre-trained parameters. For example, "ImageNet → Cars" means the transfer of the fine-tuning trajectory on the Cars dataset between the parameters pre-trained on ImageNet.

explained below, under the following two scenarios: (1) transferring learning trajectories starting from randomly initialized parameters and (2) transferring learning trajectories starting from pre-trained parameters (Section 4.1). Next, we evaluate how efficiently the transferred parameters can be trained in their subsequent training. (Section 4.2). Finally, we investigate what properties the transferred parameters inherit from the target initializations from viewpoints of loss landscape and generalization (Section 4.3). The details on experimental settings are provided in Appendix D.

**Baselines.** As baselines for learning transfer, we introduce two natural methods: **Naive** and **Oracle**. Both in the two baselines, we transfer a given learning trajectory $(\theta_1^0, \cdots, \theta_1^T)$ by a single permutation $\pi_{\mathtt{naive}}$ or $\pi_{\mathtt{oracle}}$, according to the problem formulation in Section 3.1. In the Naive baseline, we define $\pi_{\mathtt{naive}}$ as the identity permutation, which satisfies $\pi_{\mathtt{naive}}\theta = \theta$. In other words, the transferred parameter by Naive is simply obtained as $\theta_{2,\pi_{\mathtt{naive}}}^t = \theta_2^0 + (\theta_1^t - \theta_1^0)$. On the other hand, in the Oracle baseline, we first obtain a true parameter $\theta_2^T$ by actually training the given initial parameter $\theta_2^0$ with the same optimizer as training of $\theta_1^T$. Then we define $\pi_{\mathtt{oracle}}$ by minimizing the layer-wise $L^2$ distance between the actually trained trajectories $\theta_2^T - \theta_2^0$ and $\pi_{\mathtt{oracle}}(\theta_1^T - \theta_1^0)$, where we simply apply the coordinate descent as explained in Section 2.3. The Oracle baseline is expected to be close to the optimal solution for the learning transfer problem via permutation symmetry. Finally, note that the Oracle depends only on the initial and final parameter, and thus not on the the whole trajectory or its length $T$.

**Source trajectories.** In our experiments, as discussed in Section 3.4, we consider to transfer linear trajectories $[\theta_1^0 : \theta_1^T]$ of length $T$ rather than actual trajectories for $\theta_1^T$ due to the storage cost and instability emerging from noise. The transferred results for actual trajectories instead of linear ones can be found in Appendix E.1.

## 4.1 LEARNING TRANSFER EXPERIMENTS

Figure 5 shows the validation accuracies of the transferred parameters $\theta_{2,\pi_t}^t$ for each timestep $t = 1, \cdots, T$ during the transfer. For the baselines (Naive and Oracle), we set the $t$-th permutation $\pi_t$ by the fixed $\pi_{\mathtt{naive}}$ and $\pi_{\mathtt{oracle}}$ for every $t$. For our algorithms (GMT and FGMT), the $t$-th permutation $\pi_t$ corresponds to $\pi_s$ in Algorithm 1 and 2.

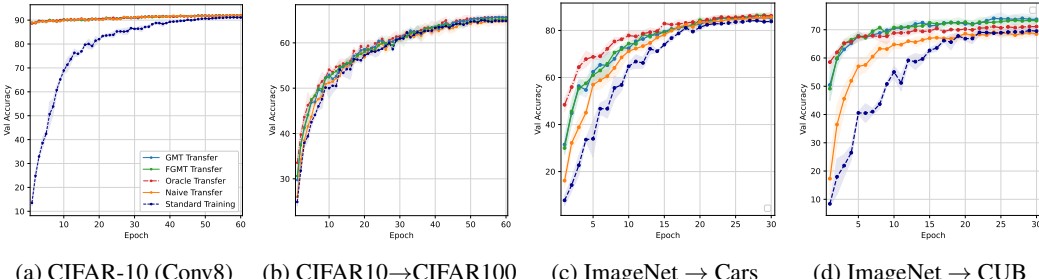

(a) CIFAR-10 (Conv8)   (b) CIFAR10→CIFAR100   (c) ImageNet → Cars   (d) ImageNet → CUB

Figure 6: Subsequent training of the transferred parameters. In each figure, we plot the validation accuracies during the training of the transferred parameters, on the same dataset as the source trajectory being trained on. The transferred parameters obtained by solving the equation (5) can be trained faster than standard training, and Naive baseline in pre-trained initialization scenario.

In the upper figures 5a-5c, we transfer a learning trajectory trained with a random initial parameter on a single dataset (MNIST (LeCun et al., 1998), CIFAR-10 (Krizhevsky, 2009) and ImageNet (Deng et al., 2009)) to another random initial parameter. We will refer to this experimental setting as the random initialization scenario. We can see that our methods successfully approximate the Oracle baseline. Also, we can see that FGMT, the fast approximation version of GMT, performs very similarly to or even outperforms GMT. This is probably because the update of $\pi_t$ affects the previously computed gradients in GMT, but not in FGMT, resulting in the stable behavior of FGMT.

In the lower figures 5f-5e, we transfer a learning trajectory of fine-tuning on a specialized dataset (a 10-classes subset of CIFAR-100 (Krizhevsky, 2009), Stanford Cars (Krause et al., 2013) and CUB-200-2011 (Wah et al., 2011)) from an initial parameter that is pre-trained on ImageNet to another pre-trained one. We refer to this experimental setting as the pre-trained initialization scenario. This scenario seems to be more difficult to transfer the learning trajectories than the random initialization scenario shown in the upper figures, since the Naive baseline always fails to transfer the trajectories. We can see that, while our methods behave closely to the Oracle baseline up to the middle of the timestep, the accuracy deteriorates immediately after that. Nevertheless, the peak accuracies of our methods largely outperform those of the Naive baseline. By stopping the transfer at the peak points (i.e., so-called early stopping), we can take an advantage of the transferred parameters as we will see in the next section.

## 4.2 ACCELERATED TRAINING OF TRANSFERRED PARAMETERS

In the previous section, we obtained the transferred parameters that achieve non-trivial accuracy without any direct training. Here we evaluate how efficiently the transferred parameters can be trained in their subsequent training, by training them on the same dataset for same epochs as the source trajectory. We started each training from the transferred parameter $\theta_{2,\pi_t}^t$ at the best trajectory step $t$ in Figure 5. Figure 6 shows the validation accuracies for each epoch in the training of the transferred parameters. In all cases, the transferred parameter can be trained faster than standard training from random/pre-trained initializations (denoted by **Standard Training**). In the random initialization scenario (6a), there seem almost no difference between four transfer methods. This is because the Naive baseline also achieves non-trivial accuracy already when transferred in this scenario. On the other hand, in the pre-trained scenarios (6b), (6c), (6d), the parameters transferred by our methods and the Oracle baseline learns the datasets faster than the parameters transferred by the Naive baseline. Thus the benefit of the transferred parameters seems to be greater especially in the pre-trained initialization scenario than in the random initialization scenario.

## 4.3 WHAT IS BEING INHERITED FROM TARGET INITIALIZATION?

In previous sections, we have observed that transferring a given learning trajectory to target initialization can achieve non-trivial accuracy beyond random guessing and indeed accelerate the subsequent training after transfer. Then the following question arises: what does the transferred parameter differ from the source parameter and inherit from the target initialization? In particular, pre-trained initializations may have their own "character" making them different from each other, due to their generalization ability or prediction mechanism.

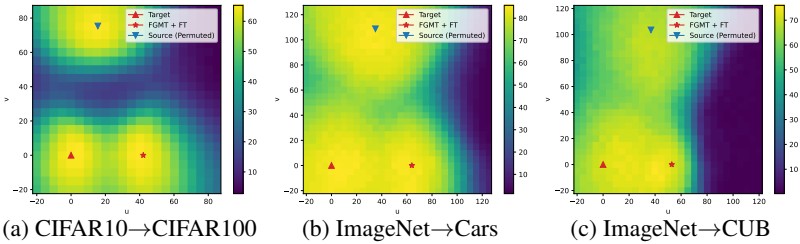

(a) CIFAR10→CIFAR100     (b) ImageNet→Cars     (c) ImageNet→CUB

Figure 7: Inheritance of basin in loss landscape of fine-tunings. We plotted the validation accuracies over the $uv$-plane following the same protocol as Garipov et al. (2018).

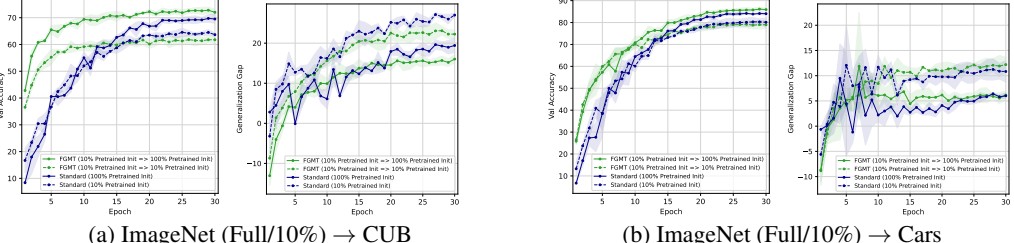

(a) ImageNet (Full/10%) → CUB         (b) ImageNet (Full/10%) → Cars

Figure 8: Inheritance of generalization ability. We plotted validation accuracies and generalization gaps (i.e., gap between training/validation accuracies) on CUB and Cars, fine-tuned from initializations pre-trained on (full or 10% subset of) ImageNet. FGMT (X ⇒ Y) means that the learning trajectory starting from an initialization X is transferred onto another initialization Y. (Even if X and Y are the same description, they are pre-trained with different random seeds. )

**Basin in loss landscape.** In context of transfer learning, Neyshabur et al. (2020) empirically showed that fine-tuning one pre-trained initialization always leads to the same basin in loss landscape, which enables the fine-tuned models to share similar properties inherited from the same pre-trained initialization. From this viewpoint, when a source trajectory with some pre-trained initialization is transferred to another (target) pre-trained initialization and then fine-tuned (referred as **FGMT+FT**), it is expected to arrive at the same basin as **Target** parameters actually fine-tuned from the target initialization. In Figure 7, we empirically validate this expectation, with comparison to the **Source (Permuted)** parameter, the end point of the source trajectory permuted with Git Re-basin (Ainsworth et al., 2023) which transforms it closest to Target. We can see that the transferred parameter lives in the nearly same basin as the target one, while there are mild barriers from the permuted source parameter. This implies that the transferred parameter inherits similar mechanism (Lubana et al., 2023) from the target initialization which cannot be obtained by simply permuting the source parameter.

**Generalization ability.** In Figure 8, we focus on the case that, unlike previous sections, the source initialization $\theta_1^0$ has different (worse) generalization ability than target initialization $\theta_2^0$. In particular, we use the source initialization $\theta_1^0$ pre-trained on 10% subset of ImageNet, and the target initialization $\theta_2^0$ pre-trained on full ImageNet. The former generalizes poorly (validation accuracy ≈ 50% on ImageNet) compared to the latter (validation accuracy ≈ 72% on ImageNet). Surprisingly, the results show that (1) learning trajectories can be transferred between pre-trained initializations even with different generalization ability, and (2) the transferred parameter successfully acquires better generalization inherited from the target initialization $\theta_2^0$, even though it has never been trained on the target initialization before/during transfer.

## 5 CONCLUSION

In this work, we formulated the problem of how we can synthesize an unknown learning trajectory from the known one, named the learning transfer problem, and derived an algorithm that approximately solves it very efficiently. In our experiments, we confirmed that our algorithm efficiently transfers a given learning trajectory to achieve non-trivial accuracy without training, and that the transferred parameters accelerate their subsequent training. Moreover, we investigated what properties the transferred parameters inherit from the target initializations from the viewpoints of loss landscape and generalization. The last observation potentially opens up new paradigm of deep learning in future: For example, when a foundation model is newly updated with better generalization or fixed vulnerability, its fine-tuned models may efficiently follow it by transferring their fine-tuning trajectory from old foundation model to the new one.

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

# Appendices

## A  VALIDATING ASSUMPTION (P) AT INITIALIZATION

In this section, we theoretically validate Assumption (P) in Section 3.3 for 2-layered ReLU neural network at initialization. Let $d$ be the dimension of inputs $x = (x_j)_{j=1,\cdots,d} \in \mathbb{R}^d$, and $f_{w,v}(x) := \sum_{i=1}^N v_i \sigma(\sum_{j=1}^d w_{ij} x_j)$ be a ReLU neural network[†] with $N$ hidden neurons, where $w = (w_{ij}) \in \mathbb{R}^{N \times d}$, $v = (v_i) \in \mathbb{R}^N$, $\sigma(z) := \max(z, 0)$.

We assume that the input $x \in \mathbb{R}^d$ and its label $y \in \mathbb{R}$ is sampled from some bounded distribution $\mathcal{D}$ whose density function $p_{\mathcal{D}}(x, y)$ is continuous over $(x, y) \in \mathbb{R}^{d+1}$. The parameters $w$ and $v$ are initialized with Kaiming uniform initialization, i.e., $w \sim U(-\frac{1}{\sqrt{d}}, \frac{1}{\sqrt{d}})^{N \times d}$, $v \sim U(-\frac{1}{\sqrt{N}}, \frac{1}{\sqrt{N}})^N$. We employ the MSE loss $\mathcal{L}(y_1, y_2) := \frac{1}{2}|y_1 - y_2|^2$ and focus on the expected gradient $\mathbb{E}_{(x,y)\sim\mathcal{D}}[\nabla_{(w,v)}\mathcal{L}(f_{w,v}(x), y)]$ for this loss function at the initialization $w, v$. Here $\nabla_{(w,v)}\mathcal{L}(f_{w,v}(x), y)$ lives in $\mathbb{R}^{N \times d} \times \mathbb{R}^d$ which is obtained as a concatenation of two vectors $(\partial\mathcal{L}(f_{w,v}(x), y)/\partial w_{ij} : i \in \{1, \cdots, N\}, j \in \{1 \cdots, d\}) \in \mathbb{R}^{N \times d}$ and $(\partial\mathcal{L}(f_{w,v}(x), y)/\partial v_i : i \in \{1, \cdots, N\}) \in \mathbb{R}^N$. Moreover, we introduce the following assumption on the data distribution $\mathcal{D}$ for technical reason:

$$\left| \mathbb{E}_{(x,y)\sim\mathcal{D}}\left[y\sigma\left(\sum_j w_{ij}x_j\right)\right]\right| \geq K \text{ for some } K > 0 \text{ with high probability w.r.t. } w_{ij}, \quad (8)$$

which is reasonable because it requires non-zero correlation between the input $x$ and its label $y$.

To validate our assumption, we introduce normalized distance $\frac{\|v_1 - v_2\|_2}{\sqrt{\|v_1\|_2\|v_2\|_2}}$ between two vectors $v_1$ and $v_2$, which can be considered as cosine distance when $\|v_1\| \approx \|v_2\|_2$ holds. Now our claim is that the normalized distance of gradients at independent random initializations can be arbitrarily small by appropriate neuron-permutation and sufficient over-parameterization. In this sense, the following theorem validates Assumption (P) at random initializations:

**Theorem A.1** *Under the above assumption, given two pairs of randomly initialized parameters $(w, v)$ and $(w', v')$, with high probability, there exists a permutation symmetry $\pi \in S_N$ such that the normalized distance between the expected gradients $\mathbb{E}_{(x,y)}[\nabla_{w,v}\mathcal{L}]$ and $\mathbb{E}_{(x,y)}[\nabla_{w'',v''}\mathcal{L}]$, where $(w'', v'')$ is the permuted parameter of $(w', v')$ with $\pi$, can be arbitrarily small when $N$ is sufficiently large.*

**Proof:**  The expected gradient at $(w, v)$ can be computed as follows:

$$
\begin{aligned}
\mathbb{E}_{(x,y)\sim\mathcal{D}}\left[\frac{\partial\mathcal{L}(f_{w,v}(x), y)}{\partial w_{ij}}\right] &= \mathbb{E}\left[(f_{w,v}(x) - y)\frac{\partial f_{w,v}(x)}{\partial w_{ij}}\right] \\
&= v_i\mathbb{E}\left[(f_{w,v}(x) - y)x_j\mathbb{1}_{\{\sum_k w_{ik}x_k > 0\}}\right], \\
\mathbb{E}_{(x,y)\sim\mathcal{D}}\left[\frac{\partial\mathcal{L}(f_{w,v}(x), y)}{\partial v_i}\right] &= \mathbb{E}\left[(f_{w,v}(x) - y)\frac{\partial f_{w,v}(x)}{\partial v_i}\right] \\
&= \mathbb{E}\left[(f_{w,v}(x) - y)\sigma\left(\sum_{j=1}^d w_{ij}x_j\right)\right] \\
&= \mathbb{E}\left[(f_{w,v}(x) - y)\left(\sum_{j=1}^d w_{ij}x_j\right)\mathbb{1}_{\{\sum_k w_{ik}x_k > 0\}}\right] \\
&= \sum_{j=1}^d w_{ij}\mathbb{E}\left[(f_{w,v}(x) - y)x_j\mathbb{1}_{\{\sum_k w_{ik}x_k > 0\}}\right]
\end{aligned}
$$

---

[†]We can ignore the non-differentiable locus of the network because we only consider expected gradients over a distribution with a continuous density function, where such low-dimensional locus has zero measure.

By Hoeffding's inequality and $v_i \sim U(-\frac{1}{\sqrt{N}}, \frac{1}{\sqrt{N}})$, with high probability, we can assume $f_{w,v}(x) \approx 0$ for sufficiently large $N$. Thus by letting $C^j(w_{i1}, \cdots, w_{id}) := \mathbb{E}\left[-yx_j \mathbb{1}_{\{\sum_k w_{ik}x_k > 0\}}\right]$, we can simplify the expected gradients as follows:

$$
\begin{aligned}
\mathbb{E}_{(x,y)\sim\mathcal{D}}\left[\frac{\partial \mathcal{L}(f_{w,v}(x), y)}{\partial w_{ij}}\right] &= v_i C^j(w_{i1}, \cdots, w_{id}), \\
\mathbb{E}_{(x,y)\sim\mathcal{D}}\left[\frac{\partial \mathcal{L}(f_{w,v}(x), y)}{\partial v_i}\right] &= \sum_{j=1}^{d} w_{ij} C^j(w_{i1}, \cdots, w_{id}),
\end{aligned}
$$

which are also valid for the counterpart $(w', v')$ instead of $(w, v)$.

Next, we take a permutation $\pi \in S_N$ by applying Lemma A.2 to $(d+1)$-dimensional random vectors $(\sqrt{d}w_{i1}, \cdots, \sqrt{d}w_{id}, \sqrt{N}v_i)$ and $(\sqrt{d}w'_{i1}, \cdots, \sqrt{d}w'_{id}, \sqrt{N}v'_i) \sim U([-1,1]^{d+1})$ where $i = 1, \cdots, N$. Let $w'' := (w'_{\pi(i)j})_{ij}$ and $v'' := (v'_{\pi(i)})_i$. Then we want to evaluate the squared normalized distance between $\mathbb{E}[\nabla_{(w,v)}\mathcal{L}]$ and $\mathbb{E}[\nabla_{(w'',v'')}\mathcal{L}]$:

$$
\frac{\left\|\mathbb{E}[\nabla_{(w,v)}\mathcal{L}] - \mathbb{E}[\nabla_{(w'',v'')}\mathcal{L}]\right\|_2^2}{\left\|\mathbb{E}[\nabla_{(w,v)}\mathcal{L}]\right\|_2 \left\|\mathbb{E}[\nabla_{(w'',v'')}\mathcal{L}]\right\|_2} \tag{9}
$$

We can evaluate each factor $\left\|\mathbb{E}[\nabla_{(w,v)}\mathcal{L}] - \mathbb{E}[\nabla_{(w'',v'')}\mathcal{L}]\right\|_2^2$, $\left\|\mathbb{E}[\nabla_{(w,v)}\mathcal{L}]\right\|_2$, $\left\|\mathbb{E}[\nabla_{(w',v')}\mathcal{L}]\right\|_2$ in Eq (9) as follows:

Claim (1): $\left\|\mathbb{E}[\nabla_{(w,v)}\mathcal{L}] - \mathbb{E}[\nabla_{(w'',v'')}\mathcal{L}]\right\|_2^2 = O(N\varepsilon^2) + o(N)$ with high probability.

Claim (2): $\left\|\mathbb{E}[\nabla_{(w,v)}\mathcal{L}]\right\|_2^2 = \Omega(N)$, $\left\|\mathbb{E}[\nabla_{(w',v')}\mathcal{L}]\right\|_2^2 = \Omega(N)$ with high probability.

Proof of Claim (1): Let $I := \{i \in \{1, \cdots, N\} : |w_{ij} - w''_{ij}| \le \sqrt{d}\varepsilon, |v_i - v''_i| \le \sqrt{N}\varepsilon\}$. We have

$$
\begin{aligned}
\left\|\mathbb{E}[\nabla_{(w,v)}\mathcal{L}] - \mathbb{E}[\nabla_{(w'',v'')}\mathcal{L}]\right\|_2^2 &= \sum_{i,j}\left|v_i C^j(w_{i*}) - v''_i C^j(w''_{i*})\right|^2 \\
&\quad + \sum_i \left|\sum_j w_{ij} C^j(w_{i*}) - w''_{ij} C^j(w''_{i*})\right|^2 \\
&\le \sum_{i=1}^{N}\sum_{j=1}^{d}\left(\left|v_i - v''_i\right| \cdot \left|C^j(w_{i*})\right| + \left|v''_i\right| \cdot \left|C^j(w_{i*}) - C^j(w''_{i*}))\right|\right)^2 \\
&\quad + \sum_i \left|\sum_j w_{ij} C^j(w_{i*}) - w''_{ij} C^j(w''_{i*})\right|^2 \\
&\le \sum_{i\in I}\sum_{j=1}^{d}\left(\left|v_i - v''_i\right| \cdot \left|C^j(w_{i*})\right| + \left|v''_i\right| \cdot \left|C^j(w_{i*}) - C^j(w''_{i*}))\right|\right)^2 \\
&\quad + \sum_{i\in I} \left|\sum_j w_{ij} C^j(w_{i*}) - w''_{ij} C^j(w''_{i*})\right|^2 \\
&\quad + O(|I^c|), \tag{10}
\end{aligned}
$$

where $I^c$ stands for the complement $\{1, \cdots, N\} \setminus I$. By Lemma A.2, the size of the complement $|I^c|$ is upper bounded by $O(\sqrt{N})$. Also we note that $C^j$ is bounded by some constant since $\mathcal{D}$ is bounded, and $|v_j|, |v''_j| \le 1/\sqrt{N}, |w_{ij}|, |w''_{ij}| \le 1/\sqrt{d}$. Combining these facts and Lemma A.3 for the second term of (10), it follows that (10) $\le O(N\varepsilon^2) + O(\sqrt{N})$.

Proof of Claim (2):

$$
\begin{aligned}
\left\| \mathbb{E}_{(x,y)\sim\mathcal{D}}[\nabla_{(w,v)}\mathcal{L}] \right\|_2^2 \; &\geq \; \sum_{i=1}^{N} \left| \mathbb{E}\Big[\frac{\partial\mathcal{L}}{\partial v_i}\Big] \right|^2 \\
&\approx \; \sum_{i=1}^{N} \left| \mathbb{E}_{(x,y)\sim\mathcal{D}}\Big[y\sigma\big(\sum_{j=1}^{d} w_{ij}x_j\big)\Big] \right|^2 \\
&\geq \; KN.
\end{aligned}
$$

Here we used $f_{w,v}(x) \approx 0$ as explained above and the assumption Eq (8) on distribution $\mathcal{D}$.

Finally, by combining Claim (1) and (2), we obtain that the normalized distance (Eq 9) converges to $C_1\varepsilon^2$ when $N \to \infty$, with some constant $C_1 > 0$, which can be arbitrarily small by controlling $\varepsilon$. $\square$

**Lemma A.2** *Let $\varepsilon > 0$ be any small real number. Let $z_1, \cdots, z_N, z_1', \cdots, z_N', \sim U([-K,K]^k)$ be i.i.d. uniform random variables where $K > 0$ is a fixed constant. When $N$ is sufficiently large, with high probability, there exists a permutation $\pi \in S^N$ such that the number of indices $i \in \{1, \cdots, N\}$ satisfying $|z_{ij} - z_{\pi(i)j}'| > \varepsilon$ for some $j$ is upper bounded by $O(\sqrt{N})$.*

**Proof:** We can assume $K = 1$ without loss of generality by re-scaling $z_i$, $z_i'$ and $\varepsilon$. Here we follow the argument given in Entezari et al. (2021). For simplicity, we take $M \in \mathbb{N}$ satisfying $\frac{1}{M} \leq \varepsilon \leq \frac{1}{M-1}$. For each $\mathbf{l} = (l_1, \cdots, l_k) \in L := \{1, \cdots, 2M\}^k$, we consider

$$
Q_{\mathbf{l}} := (-1 + \frac{l_1 - 1}{M}, -1 + \frac{l_1}{M}) \times \cdots \times (-1 + \frac{l_k - 1}{M}, -1 + \frac{l_k}{M}),
$$

$$
n_{\mathbf{l}} := \#\{i \in I : z_i \in Q_{\mathbf{l}}\}, \quad n_{\mathbf{l}}' := \#\{i \in I : z_i' \in Q_{\mathbf{l}}\}.
$$

Note that any $z_i$ and $z_i'$ are contained in $Q_{\mathbf{l}}$ for some $\mathbf{l} \in L$ with probability 1. For each $\mathbf{l} \in L$, the number $n_{\mathbf{l}}$ and $n_{\mathbf{l}}'$ can be considered as the sum of random binary variables $b_1 + \cdots + b_N$ where each $b_j$ is sampled from Bernoulli distribution $\mathrm{Ber}(\frac{1}{M})$. By Hoeffding's inequality for the Bernoulli random variables, we have

$$
\mathbf{P}\left( \left| n_{\mathbf{l}} - \frac{N}{M} \right| \geq t \right) \leq 2\exp\left( -\frac{t^2}{2N} \right)
$$

for each $\mathbf{l} \in L$. Thus, with probability $1 - \delta$, we obtain

$$
\left| n_{\mathbf{l}} - \frac{N}{M} \right|, \left| n_{\mathbf{l}}' - \frac{N}{M} \right| \geq \sqrt{2N\log\left(\frac{4|L|}{\delta}\right)}.
$$

To construct the desired correspondence $\pi$ between $(z_i : 1 \leq i \leq N)$ and $(z_j' : 1 \leq j \leq N)$, each $z_i \in Q_{\mathbf{l}}$ should be mapped to some $z_j' \in Q_{\mathbf{l}}$. The number of $\{i \in \{1, \cdots, N\} : z_i$ does not have its counterpart $z_j'\}$ can be upper bounded by

$$
\sum_{\mathbf{l} \in L} |n_{\mathbf{l}} - n_{\mathbf{l}}'| \leq \sum_{\mathbf{l} \in L} \left| n_{\mathbf{l}} - \frac{N}{M} \right| + \left| n_{\mathbf{l}}' - \frac{N}{M} \right| \leq 2|L|\sqrt{2N\log\left(\frac{4|L|}{\delta}\right)} = O(\sqrt{N}).
$$

$\square$

**Lemma A.3** *Assume that $(w_1, \cdots, w_d), (w_1', \cdots, w_d') \in \left[-\frac{1}{\sqrt{d}}, \frac{1}{\sqrt{d}}\right]^d$ satisfy $\sup_k |w_k - w_k'| \leq \varepsilon$. It follows that $\left| \sum_j w_j C^j(w_1, \cdots, w_d) - \sum_j w_j' C^j(w_1', \cdots, w_d') \right| \leq O(\varepsilon)$.*

**Proof:** Let $G(w_1, \cdots, w_d) := \sum_j w_j C^j(w_1, \cdots, w_d) = \mathbb{E}[-y\sigma(\sum_j w_j x_j)]$. By applying triangle inequality iteratively, we have

$$
\begin{aligned}
&|G(w_1, \cdots, w_d) - G(w_1', \cdots, w_d')| \\
\leq\; &|G(w_1, \cdots, w_d) - G(w_1', w_2, \cdots, w_d)| + |G(w_1', w_2, \cdots, w_d) - G(w_1', \cdots, w_d')| \\
\leq\; &\cdots \\
\leq\; &|G(w_1, \cdots, w_d) - G(w_1', w_2, \cdots, w_d)| + \cdots + |G(w_1', \cdots, w_d', w_d) - G(w_1', \cdots, w_d')|
\end{aligned}
$$

Thus the proof can be reduced to the case where $|w_{k_0} - w'_{k_0}| \leq \varepsilon$ for some $k_0$ and $w_j = w'_j$ for $k \neq k_0$. We can assume $k_0 = 1$ and $w_1 \leq w'_1 \leq w_1 + \varepsilon$ without loss of generality. Then we have

$$
\begin{aligned}
&\left| G(w_1, \cdots, w_d) - G(w'_1, \cdots, w'_d) \right| \\
=\ &\left| \mathbb{E}\Big[ -y\sigma\Big( \sum_j w_j x_j \Big) \Big] - \mathbb{E}\Big[ -y\sigma\Big( \sum_j w'_j x_j \Big) \Big] \right| \\
=\ &\left| \mathbb{E}\Big[ y\Big\{ \sigma\Big( \sum_j w_j x_j \Big) - \sigma\Big( \sum_j w'_j x_j \Big) \Big\} \Big] \right| \\
\leq\ &\int |y| \cdot \left| \Big\{ \sigma\Big( w_1 x_1 + \sum_{j=2}^{d} w_j x_j \Big) - \sigma\Big( w'_1 x_1 + \sum_{j=2}^{d} w_j x_j \Big) \Big\} \right| \cdot p(x,y) dx dy \\
\leq\ &\int |y| \cdot \left| w_1 x_1 + \sum_{j=2}^{d} w_j x_j \right| \cdot \mathbb{1}_{\{w_1 x_1 + \sum_{j=2}^{d} w_j x_j \geq 0, w'_1 x_1 + \sum_{j=2}^{d} w_j x_j < 0\}} \cdot p(x,y) dx dy \\
&+ \int |y| \cdot \left| w'_1 x_1 + \sum_{j=2}^{d} w_j x_j \right| \cdot \mathbb{1}_{\{w_1 x_1 + \sum_{j=2}^{d} w_j x_j < 0, w'_1 x_1 + \sum_{j=2}^{d} w_j x_j \geq 0\}} \cdot p(x,y) dx dy \\
&+ \int |y| \cdot \left| \Big( w_1 x_1 + \sum_{j=2}^{d} w_j x_j \Big) - \Big( w'_1 x_1 + \sum_{j=2}^{d} w_j x_j \Big) \right| \cdot p(x,y) dx dy
\end{aligned}
\tag{11}
$$

On the first term of the last inequality (11), since $w'_1 x_1 + \sum_{j=2}^{d} w_j x_j < 0$ holds on the integrated region, it follows that $0 \leq w_1 x_1 + \sum_{j=2}^{d} w_j x_j < w_1 x_1 - w'_1 x_1$. Thus we have

$$
\begin{aligned}
&\int |y| \cdot \left| w_1 x_1 + \sum_{j=2}^{d} w_j x_j \right| \cdot \mathbb{1}_{\{w_1 x_1 + \sum_{j=2}^{d} w_j x_j \geq 0, w'_1 x_1 + \sum_{j=2}^{d} w_j x_j < 0\}} \cdot p(x,y) dx dy \\
\leq\ &\int |y| \cdot |w_1 x_1 - w'_1 x_1| \cdot p(x,y) dx dy \leq \int |y| \cdot \varepsilon |x_1| \cdot p(x,y) dx dy
\end{aligned}
$$

The same argument holds for the second term of inequality (11). Furthermore, the third term is also bounded by $\int |y| \cdot \varepsilon |x_1| \cdot p(x,y) dx dy$. Therefore, by the boundedness of the distribution $\mathcal{D}$, we have (11) $\leq 3\varepsilon \int |yx_1| p(x,y) dx dy = O(\varepsilon)$. $\qquad \square$

## B  Proof of Lemma 3.2

In this section, we employ the same notation as Section 3.3. We assume that

1. $\|(\theta_i^{t+1} - \theta_i^t) - (-\alpha_t \nabla_{\theta_i^t} \mathcal{L})\|_2 < \varepsilon$,

2. $\|\pi_s \nabla_{\theta_1^t} \mathcal{L} - \nabla_{\theta_{2,\pi_{s-1}}^t} \mathcal{L}\|_2 < \varepsilon$, for $t \leq s - 1$,

3. The gradient $\nabla_\theta \mathcal{L}$ is $K$-Lipschitz continuous with respect to the parameter $\theta$, i.e., $\|\nabla_\theta \mathcal{L} - \nabla_{\theta'} \mathcal{L}\|_2 \leq K\|\theta - \theta'\|_2$.

**Lemma B.1** *Under the above assumptions, we have*

$$
\theta_{2,\pi_{s'}}^t - \theta_{2,\pi_s}^t = O(T^{s'} K^{s'} \varepsilon),
\tag{12}
$$

*for $0 \leq t \leq s < s' \leq T$.*

**Proof:** We prove by induction:

$$
\begin{aligned}
\theta_{2,\pi_{s'}}^t &= \theta_2^0 + \pi_{s'}(\theta_1^t - \theta_1^0) \\
&= \theta_2^0 + \pi_{s'}\left(-\sum_{t'=0}^{t-1}\alpha_{t'}\nabla_{\theta_1^{t'}}\mathcal{L} + O(t\varepsilon)\right) && \text{(by Assumption 1.)} \\
&= \theta_2^0 + \sum_{t'=0}^{t-1}\left(-\alpha_{t'}\pi_{s'}\nabla_{\theta_1^{t'}}\mathcal{L}\right) + O(t\varepsilon) \\
&= \theta_2^0 + \sum_{t'=0}^{t-1}\left(-\alpha_{t'}\nabla_{\theta_{2,\pi_{s'-1}}^{t'}}\mathcal{L} + O(\varepsilon)\right) + O(t\varepsilon) && \text{(by Assumption 2.)} \\
&= \theta_2^0 + \sum_{t'=0}^{t-1}\left(-\alpha_{t'}\nabla_{\theta_{2,\pi_{s-1}}^{t'}+O(T^{s'-1}K^{s'-1}\varepsilon)}\mathcal{L} + O(\varepsilon)\right) + O(t\varepsilon) && \text{(by induction hypothesis.)} \\
&= \theta_2^0 + \sum_{t'=0}^{t-1}\left(-\alpha_{t'}\nabla_{\theta_{2,\pi_{s-1}}^{t'}}\mathcal{L}\right) + O(T^{s'}K^{s'}\varepsilon) && \text{(by Assumption 3.)} \\
&= \theta_2^0 + \pi_s\left(-\sum_{t'=0}^{t-1}\alpha_{t'}\nabla_{\theta_1^{t'}}\mathcal{L}\right) + O(T^{s'}K^{s'}\varepsilon) \\
&= \theta_2^0 + \pi_s\left(\theta_1^t - \theta_1^0\right) + O(T^{s'}K^{s'}\varepsilon) \\
&= \theta_{2,\pi_s}^t + O(T^{s'}K^{s'}\varepsilon).
\end{aligned}
$$

$\square$

## C RELATED WORK

**Loss landscape, linear mode connectivity, permutation symmetry.** Loss landscape of training deep neural network has been actively studied in an effort to unravel mysteries of non-convex optimization in deep learning (Hochreiter & Schmidhuber, 1997; Choromanska et al., 2015; Lee et al., 2016; Keskar et al., 2017; Li et al., 2018). One of the mysteries in deep learning is the stability and consistency of their training processes and solutions, despite of the multiple sources of randomness such as random initialization, data ordering and data augmentation (Fort et al., 2019; Bhojanapalli et al., 2021; Summers & Dinneen, 2021; Jordan, 2023). Previous studies of mode connectivity both theoretically (Freeman & Bruna, 2017; Simsek et al., 2021) and empirically (Draxler et al., 2018; Garipov et al., 2018) demonstrate the existence of low-loss curves between any two optimal solutions trained independently with different randomness.

Linear mode connectivity (LMC) is a special case of mode connectivity where two optimal solutions are connected by a low-loss linear path (Nagarajan & Kolter, 2019; Frankle et al., 2020; Mirzadeh et al., 2021; Entezari et al., 2021; Benzing et al., 2022; Ainsworth et al., 2023; Juneja et al., 2023; Lubana et al., 2023). In this line of research, Entezari et al. (2021) observed that even two solutions trained from different random initialization can be linearly connected by an appropriate permutation symmetry. Ainsworth et al. (2023) developed an efficient method to find such permutations, and Jordan et al. (2023) extends it to NN architectures with Batch normalization (Ioffe & Szegedy, 2015). Their observations strength the expectation on some sort of similarity between two training processes even from different random initializations, via permutation symmetry. In our work, based on these observations, we attempt to transfer one training process to another initial parameter by permutation symmetry.

Another line of research related to our work is the studies of monotonic linear interpolation (MLI) between an initialization and its trained result. Goodfellow et al. (2015) first observed that the losses are monotonically decreasing along the linear path between an initial parameter and the trained one. Frankle (2020) and Lucas et al. (2021) confirmed that the losses are monotonically non-increasing even with modern network architectures such as CNNs and ResNets (He et al., 2016). Vlaar & Frankle (2022) empirically analyzed which factor in NN training influences the shape of the non-increasing

loss curve along the linear interpolation, and Wang et al. (2023) theoretically analyzed the plateau phenomenon in the early phase of the linear interpolation. Motivated by these observations, we introduced the notion of linear trajectories in Section 3.4 to reduce storage costs in our learning transfer.

**Model editing.** Our approach of transferring learning trajectories can be also considered as a kind of model editing (Sinitsin et al., 2020; Santurkar et al., 2021; Ilharco et al., 2022; 2023) in the parameter space because we modify a given initial parameter by adding an appropriately permuted trajectory. In particular, a recent work by Ilharco et al. (2023) is closely related to our work. They proposed to arithmetically edit a pre-trained NN with a task vector, which is defined by subtracting the initial pre-trained parameter from the parameter fine-tuned on a specific task. From our viewpoint, task vectors can be seen as one-step learning trajectories (i.e., learning trajectories with $T = 1$). Model merging (or model fusion) (Singh & Jaggi, 2020; Matena & Raffel, 2022; Wortsman et al., 2022; Li et al., 2023) is also related in the sense of the calculation in the parameter space.

**Efficient training for multiple NNs.** There are several literatures that attempt to reduce the computation costs in training multiple NNs. Fast ensemble is an approach to reduce the cost in ensemble training by cyclically scheduled learning rate (Huang et al., 2017) or by searching different optimal basins in loss landscape (Garipov et al., 2018; Fort et al., 2019; Wortsman et al., 2021; Benton et al., 2021). A recent work by Liu et al. (2022) leverages knowledge distillation (Hinton et al., 2015) from one training to accelerate the subsequent trainings. Our approach differs from theirs in that we try to establish a general principle to transfer learning trajectories. Also, the warm-starting technique investigated by Ash & Adams (2020) seems to be related in that they subsequently train from a once trained network. There may be some connection between their and our approaches, which remains for future work.

**Gradient matching.** The gradient information obtained during training has been utilized in other areas outside of ours. For example, in dataset distillation, Zhao et al. (2021) optimized a distilled dataset by minimizing layer-wise cosine similarities between gradients on the distilled dataset and the real one, starting from random initial parameters, which leads to similar training results on those datasets. Similarly, Yin et al. (2021) successfully recovered private training data from its gradient by minimizing the distance between gradients. In contrast to their problem where input data is optimized, our problem requires optimizing unknown transformation for NN parameters. In addition, our problem requires matching the entire learning trajectories, which are too computationally expensive to be computed naively.

## D DETAILS FOR OUR EXPERIMENTS

### D.1 DATASETS

The datasets used in our experiments (Section 4) are listed below. For all datasets, we split the officially given training dataset into 9:1 for training and validation.

- **MNIST.** MNIST (LeCun et al., 1998) is a dataset of $28 \times 28$ images of hand-written digits, which is available under the terms of the CC BY-SA 3.0 license.
- **CIFAR-10, CIFAR100.** CIFAR-10 and CIFAR-100 (Krizhevsky, 2009) are datasets of $32 \times 32$ images with 10 and 100 classes respectively.
- **ImageNet.** ImageNet (Deng et al., 2009) is a large-scale dataset of images with 1000 classes, which is provided for non-commercial research or educational use.
- **Stanford Cars.** Stanford Cars (Krause et al., 2013) is a dataset of images with 196 classes of cars, which is provided for research purposes. We refer to this dataset as Cars for short.
- **CUB-200-2011.** CUB-200-2011 (Wah et al., 2011) is a dataset of images of 200 species of birds. We refer to this dataset as CUB for shot.

### D.2 NETWORK ARCHITECTURES

The neural network architectures used in our experiments (Section 4) are listed as follows:

- **2-MLP.** 2-MLP is a two-layered neural network with the ReLU activations. The design of this architecture is shown in Table 1.
- **Conv8.** Conv8 is an 8-layered CNN followed by three linear and ReLU layers. The design of this architecture is shown in Table 2.
- **ResNet-18.** The ResNet family (He et al., 2016) is a series of deep CNNs with skip connections. We employed the standard 18-layered one for ResNet-18.

Table 1: The architecture of 2-MLP.

| No. | Layers | Output dimensions |
| --- | --- | --- |
| 1 | Flattening | $784 \, (= 28 \times 28)$ |
| 2 | Linear $\rightarrow$ ReLU | 4096 |
| 4 | Linear | 10 |
| 5 | Softmax | 10 |

Table 2: The architecture of Conv8.

| No. | Layers |
| --- | --- |
| 1 | Conv(input=3, output=64, kernel_size=$(3, 3)$, stride=1, padding=1) $\rightarrow$ ReLU |
| 2 | Conv(input=64, output=64, kernel_size=$(3, 3)$, stride=1, padding=1) $\rightarrow$ ReLU |
| 3 | MaxPooling(kernel_size=$(2, 2)$) |
| 4 | Conv(input=64, output=128, kernel_size=$(3, 3)$, stride=1, padding=1) $\rightarrow$ ReLU |
| 5 | Conv(input=128, output=128, kernel_size=$(3, 3)$, stride=1, padding=1) $\rightarrow$ ReLU |
| 6 | MaxPooling(kernel_size=$(2, 2)$) |
| 7 | Conv(input=128, output=256, kernel_size=$(3, 3)$, stride=1, padding=1) $\rightarrow$ ReLU |
| 8 | Conv(input=256, output=256, kernel_size=$(3, 3)$, stride=1, padding=1) $\rightarrow$ ReLU |
| 9 | MaxPooling(kernel_size=$(2, 2)$) |
| 10 | Conv(input=256, output=512, kernel_size=$(3, 3)$, stride=1, padding=1) $\rightarrow$ ReLU |
| 11 | Conv(input=512, output=512, kernel_size=$(3, 3)$, stride=1, padding=1) $\rightarrow$ ReLU |
| 12 | MaxPooling(kernel_size=$(4, 4)$) |
| 13 | Conv(input=512, output=512, kernel_size=$(3, 3)$, stride=1, padding=1) $\rightarrow$ ReLU |
| 14 | Linear(input=512, output=256) $\rightarrow$ ReLU |
| 15 | Linear(input=256, output=256) $\rightarrow$ ReLU |
| 16 | Linear(input=512, output=10) |
| 17 | Softmax |

## D.3 TRAINING DETAILS

### D.3.1 DETAILS ON IMPLEMENTATION AND DEVICES

We implemented the codebase for all experiments in Python 3 with the PyTorch library (Paszke et al., 2019). Our computing environment is a machine with 12 Intel CPUs, 140 GB CPU memory and a single A100 GPU.

### D.3.2 TRAINING OF SOURCE TRAJECTORIES

In training for source trajectories, we used SGD with momentum in PyTorch for the optimization. It has the following hyperparameters: the total epoch number $E$, batch size $B$, learning rate $\alpha$, weight decay $\lambda$, momentum coefficient $\mu$. We used the cosine annealing (Loshchilov & Hutter, 2017) for scheduling the learning rate $\eta$ except for MNIST. For the random initialization, we used the standard Kaiming initialization (He et al., 2015), which is also a default in PyTorch. The details on the hyperparameters are as follows:

- **2-MLP on MNIST.** We used $E = 15$, $B = 128$, $\alpha = 0.01$, $\lambda = 0.0$, $\mu = 0.9$.
- **Conv8 on CIFAR-10.** We used $E = 60$, $B = 128$, $\alpha = 0.05$, $\lambda = 0.0001$, $\mu = 0.9$.

- **Conv8 on CIFAR-100.** We used $E = 30$, $B = 128$, $\alpha = 0.05$, $\lambda = 0.0001$, $\mu = 0.9$, starting from the pre-trained parameter on CIFAR-10.
- **ResNet18 on ImageNet.** We used $E = 100$, $B = 128$, $\alpha = 0.1$, $\lambda = 0.0001$, $\mu = 0.9$. For the first 5 epochs, we gradually increased the learning rate as $\eta = 0.1 \times (i/5)$ for each $i$-th epoch ($i = 1, \cdots, 5$). For the last 95 epochs, we decayed the learning rate by cosine annealing starting from $\eta = 0.1$.
- **ResNet18 on Cars.** We used $E = 30$, $B = 128$, $\alpha = 0.1$, $\lambda = 0.0001$, $\mu = 0.9$, starting from the pre-trained parameter on ImageNet.
- **ResNet18 on CUB.** We used $E = 30$, $B = 128$, $\alpha = 0.1$, $\lambda = 0.0001$, $\mu = 0.9$, starting from the pre-trained parameter on ImageNet.

### D.3.3 HYPERPARAMETERS FOR TRANSFERRING LEARNING TRAJECTORIES

Our methods (Algorithm 1, 2) have the following hyperparameters: the length $T$ of learning trajectories, the batch size $B$ for each gradient matching. Also, for NN architectures with the Batch normalization (Ioffe & Szegedy, 2015) such as ResNets, the batch size $B'$ for resetting the means and variances in the Batch Normalization layers (Jordan et al., 2023) is also a hyperparameter.

- **2-MLP on MNIST.** We used $B = 128$ and $T = 5$.
- **Conv8 on CIFAR-10.** We used $B = 128 \times 2$ and $T = 30$.
- **Conv8 on CIFAR-100.** We used $B = 128 \times 2$ and $T = 15$.
- **ResNet18 on ImageNet.** We used $B = 128 \times 100$, $B' = 128 \times 20$ and $T = 40$.
- **ResNet18 on Cars.** We used $B = 128 \times 5$, $B' = 128 \times 2$ and $T = 15$.
- **ResNet18 on CUB.** We used $B = 128 \times 5$, $B' = 128 \times 2$ and $T = 15$.

### D.3.4 COMPUTATIONAL COST OF GMT/FGMT

Total computational cost of FGMT (resp. GMT) consists of: computing $2BT$ (resp. $2BT^2$) gradients, which is the same budget as $2T$ iterations of standard training, and solving lightweight[‡] optimizations (eq. 6) for $T$ times. Thus, for FGMT, the required computational cost should be significantly smaller than standard training (if implemented efficiently). We also provide the wall-clock time in our experiments in Appendix E.6, although the results are worse than the above estimate due to inefficiency of our implementation, non-negligible I/O delays and etc.

### D.3.5 SUBSEQUENT TRAINING OF TRANSFERRED PARAMETERS

For the subsequent training in Section 4.2, we used the same optimizer and learning rate scheduler as training of source trajectories, with slightly small initial learning rates ($\alpha = 0.01$ on CIFAR-10, $\alpha = 0.05$ on CIFAR-100, $\alpha = 0.05$ on Cars, $\alpha = 0.01$ for CUB), which are selected based on the validation accuracy of Naive baseline for fair comparison.

---

[‡]depends on model architectures, typically 1 or 2 seconds for each optimization in our environment

# E    ADDITIONAL EXPERIMENTAL RESULTS

## E.1    LEARNING TRANSFER FOR ACTUAL TRAJECTORIES

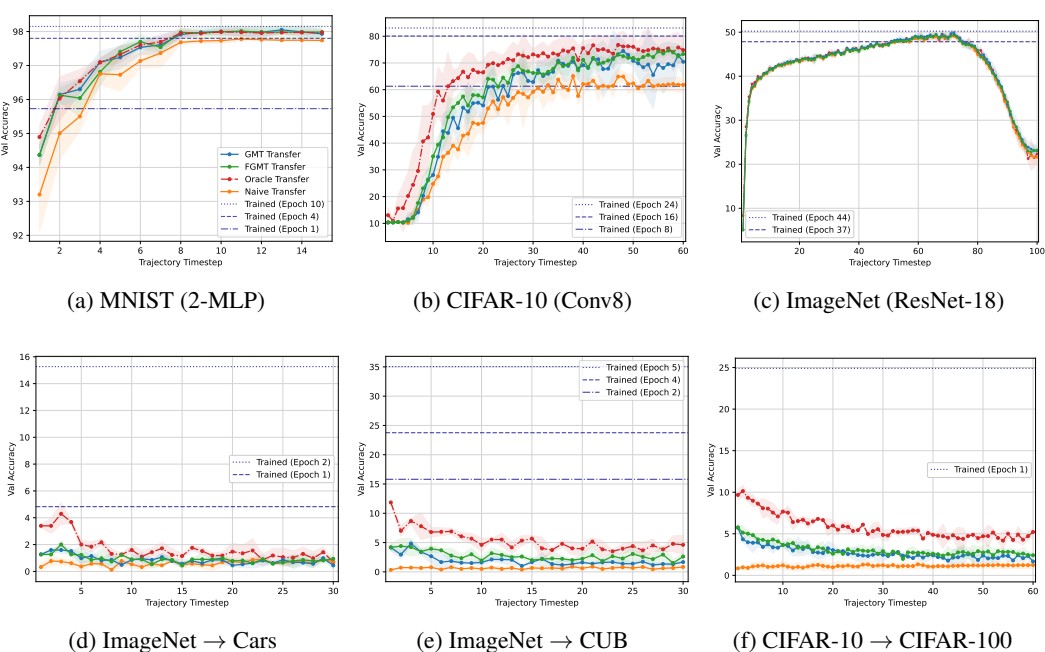

(a) MNIST (2-MLP)              (b) CIFAR-10 (Conv8)              (c) ImageNet (ResNet-18)

(d) ImageNet → Cars          (e) ImageNet → CUB              (f) CIFAR-10 → CIFAR-100

Figure 9: We plot the validation accuracies of the transferred parameter $\theta_{2,\pi_t}^t$ for each timestep $t = 1, \cdots, T$ with various datasets and NN architectures as in Figure 5, except that the actual trajectory $\theta_1^t$ at $t$-th epoch of SGD is transferred instead of linear trajectories. Compared to the results for linear trajectories in Figure 5, the transferred results for actual trajectories tend to have more variance in validation accuracy and fail to transfer in fine-tuning scenario.

## E.2    LEARNING TRANSFER ON SUN397 AND iNATURALIST2017

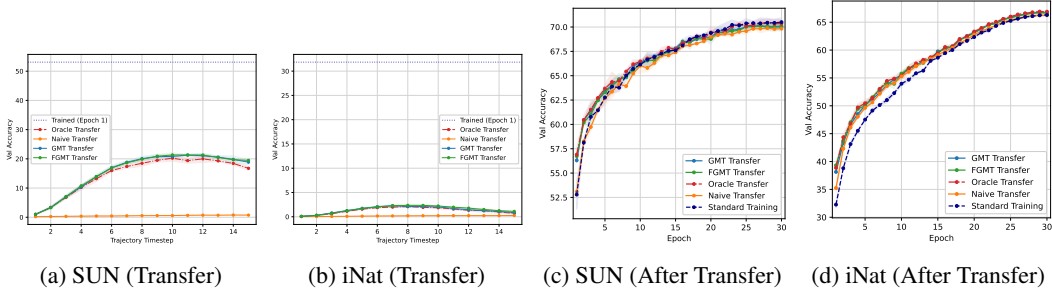

(a) SUN (Transfer)      (b) iNat (Transfer)      (c) SUN (After Transfer)      (d) iNat (After Transfer)

Figure 10: Experiments on large-scale fine-grained datasets, SUN-397 (SUN, Xiao et al. (2010)) and iNaturalist-2017 (iNat, Cui et al. (2018)) with ResNet-18, transferring fine-tuning trajectories on each dataset starting from ImageNet pre-trained initializations. Figs. (a, b) show validation accuracies during transfer, and Figs. (c, d) show the results of subsequent training after transfer, as in Section 4. Although validation accuracies during transfer are worse than standard results obtained for 1 epoch, we observed that it still accelerates the subsequent training a bit. In practice, our method still needs to be improved for such large-scale complex datasets, but the results indicate the potential effectiveness of learning transfer for such datasets.

## E.3    LEARNING TRANSFER WITH RESNET-34

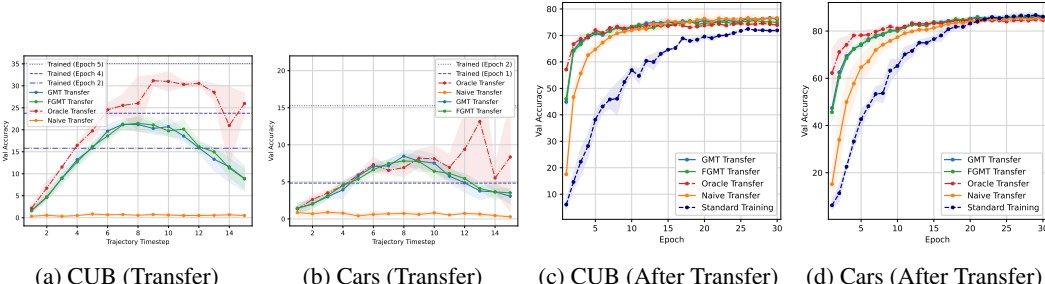

(a) CUB (Transfer)      (b) Cars (Transfer)      (c) CUB (After Transfer)      (d) Cars (After Transfer)

Figure 11: Experiments with ResNet-34. Figs. (a, b) show validation accuracies during transfer, and Figs. (c, d) show the results of subsequent training after transfer, as in Section 4. We can see that the overall trends are similar to the results with ResNet-18.

## E.4    LEARNING TRANSFER WITH ADAM

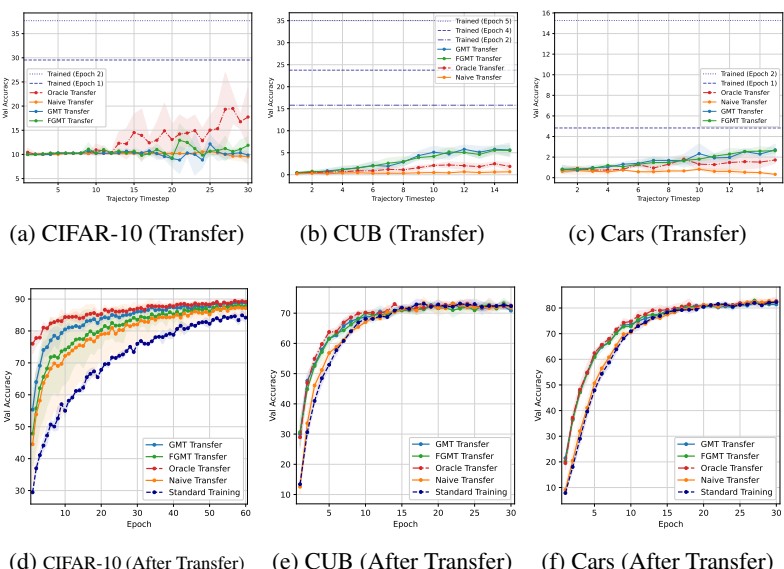

(a) CIFAR-10 (Transfer)      (b) CUB (Transfer)      (c) Cars (Transfer)

(d) CIFAR-10 (After Transfer)      (e) CUB (After Transfer)      (f) Cars (After Transfer)

Figure 12: Experiments of transferring learning trajectories obtained by Adam optimizer (Kingma & Ba, 2015). In Figs. (a-c), the validation accuracies during transfer are terribly worse than the case of SGD in Section 4.1. This may be because optimization trajectories of Adam are more complex than SGD as known in studies of monotonic linear interpolation (Lucas et al., 2021), where MLI fails in many cases with Adam. However, in Figs. (d-f), we observed that the acceleration in subsequent training after transfer (Section 4.2) still works a bit even in these cases. Thus, although our proposed method should be improved for Adam, the results indicate the potential effectiveness of learning transfer even for Adam optimization.

## E.5    ENSEMBLE EVALUATION

|  | GMT | FGMT | Standard Ensemble |
|---|---|---|---|
| CIFAR-10 | 92.01 (@50ep) | 92.11 (@50ep) | **92.26** (@60ep) |
| CIFAR-100 | **70.41** (@50ep) | 69.53 (@50ep) | 69.81 (@60ep) |
| Cars | 86.83 (@20ep) | 86.61 (@20ep) | **87.10** (@30ep) |
| CUB | 74.71 (@10ep) | **75.62** (@10ep) | 74.02 (@30ep) |

Table 3: We evaluate ensembles of three transferred models with fewer subsequent training epochs, compared to standard ensemble of three models trained from scratch. "@ X ep" means that each member of the ensembles are trained for X epochs after transferred (in GMT/FGMT) or from scratch (in Standard Ensemble).

| | Matching Probability | | | | Calibration Error (L2) | | | |
|---|---|---|---|---|---|---|---|---|
| Datasets | FGMT | GMT | Standard Ensemble | Single | FGMT | GMT | Standard Ensemble | Single |
| CIFAR-10 | 0.96 | 0.96 | 0.94 | 1.0 | 0.09 | 0.083 | 0.082 | 0.078 |
| CIFAR-100 | 0.77 | 0.77 | 0.8 | 1.0 | 0.17 | 0.17 | 0.18 | 0.18 |
| Cars | 0.89 | 0.89 | 0.91 | 1.0 | 0.050 | 0.045 | 0.053 | 0.052 |
| CUB | 0.83 | 0.83 | 0.84 | 1.0 | 0.082 | 0.084 | 0.092 | 0.099 |

Table 4: We evaluate the diversity of the members of ensemble. Matching Probability measures the probability that a prediction from each member of the ensemble matches the ensembled prediction. From this evaluation, we can see that the diversity in the ensemble by FGMT/GMT is the same level as the standard ensemble, compared to non-ensemble one (Single). We also evaluate the calibration error, but this metric does not give us any insight in our experiments because it cannot distinguish between ensemble results (Standard Ensemble) and non-ensemble results (Single).

## E.6    WALL-CLOCK TIME FOR LEARNING TRANSFER

| | | FGMT | | | GMT | | | SGD (Reference) | |
|---|---|---|---|---|---|---|---|---|---|
| Models | Datasets | GM | GC | Total | GM | GC | Total | 1-epoch | Full epochs |
| 2-MLP | MNIST | 0.66 | 16.63 | 20.62 | 0.54 | 17.22 | 20.60 | 3.69 | 55.35 |
| Conv8 | CIFAR-10 | 44.02 | 57.73 | 132 | 57.74 | 572 | 645 | 7.66 | 459 |
| Conv8 | CIFAR-100 | 71.50 | 92.10 | 179 | 82.03 | 229 | 323 | 8.53 | 511 |
| ResNet-18 | ImageNet | 253 | 3178 | 4239 | 480 | 59053 | 60259 | 5000 | 500030 |
| ResNet-18 | SUN397 | 26.52 | 541 | 779 | 26.21 | 3694 | 3914 | 403 | 12117 |
| ResNet-18 | iNaturalist2017 | 23.4 | 974 | 1321 | 30.23 | 7000 | 7343 | 1926 | 57787 |
| ResNet-18 | Cars | 46.09 | 158 | 269 | 52.95 | 1080 | 1201 | 16.90 | 507 |
| ResNet-18 | CUB | 38.33 | 121 | 217 | 44.87 | 850 | 949 | 12.94 | 388 |
| ResNet-34 | Cars | 36.84 | 174 | 282 | 39.26 | 1669 | 1808 | 20.43 | 612 |
| ResNet-34 | CUB | 37.38 | 144 | 243 | 34.75 | 979 | 1070 | 15.71 | 471 |

Table 5: Wall-clock time (seconds) of FGMT/GMT, benchmarked on a computing node with Intel Xeon Gold 6148 CPU and V100 GPU. Since the computing node is shared with other research groups and has non-negligible I/O delay due to its distributed storage, the results are highly unreliable and should not be taken seriously. GM stands for total time of Gradient Matching (i.e., line 7 in Algorithm 1 or line 5 in Algorithm 2) during the algorithms, and GC stands for total time of Gradient Computation (i.e., lines 4-5 in Algorithm 1 or lines 3-4 in Algorithm 2). Total stands for the total time of the whole algorithms. Note that the computational complexity for GM is $O(T)$ in both FGMT and GMT, and for GC is $O(T)$ in FGMT and $O(T^2)$ in GMT.

### E.7 Ablation analysis on the trajectory length $T$

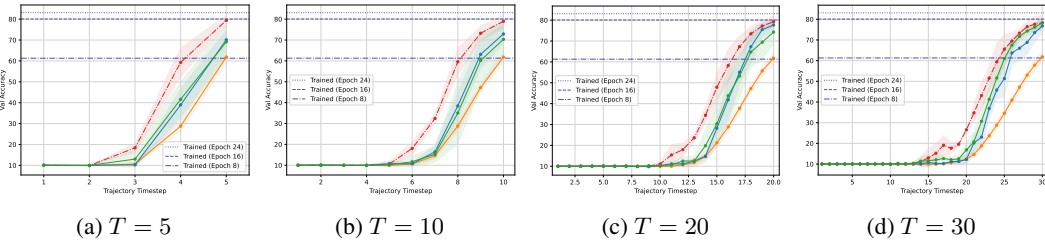

| (a) $T = 5$ | (b) $T = 10$ | (c) $T = 20$ | (d) $T = 30$ |
|---|---|---|---|

Figure 13: Ablation analysis of $T$ on CIFAR-10 with Conv8.

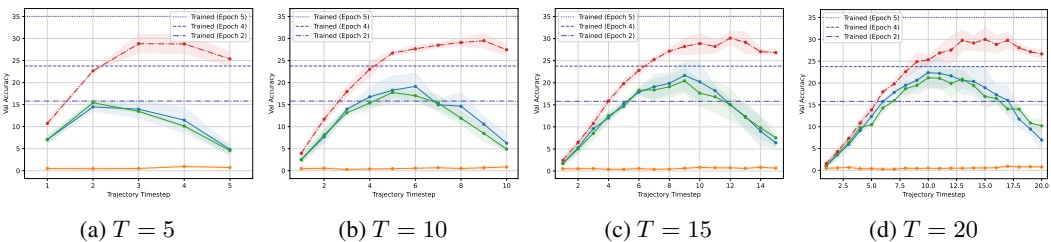

| (a) $T = 5$ | (b) $T = 10$ | (c) $T = 15$ | (d) $T = 20$ |
|---|---|---|---|

Figure 14: Ablation analysis of $T$ on CUB with ResNet18.

### E.8 Additional experiments on Assumption (P)

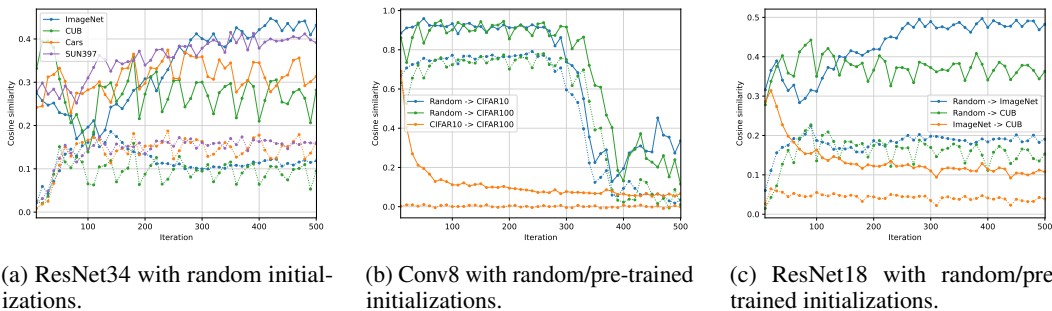

(a) ResNet34 with random initializations.

(b) Conv8 with random/pre-trained initializations.

(c) ResNet18 with random/pre-trained initializations.

Figure 15: As an addendum to Figure 3, we provide additional experiments on Assumption (P). Figure (a) shows that Assumption (P) holds weakly for ResNet34 compared to other network architectures. Figure (b, c) show how Assumption (P) behaves for random/pre-trained initializations. The label $X \rightarrow Y$ means that the initialization is random or pre-trained on $X$, and then trained on $Y$. In pre-trained initialization case, we can see that Assumption (P) only holds in the very early phase of training.

### E.9 EMPIRICAL VALIDATION FOR LEMMA 3.2

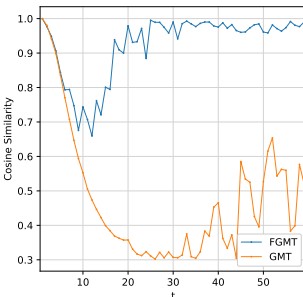

Figure 16: Here we empirically validate Lemma 3.2 on CIFAR-10 with Conv8, by observing the cosine similarity between $\theta^s_{2,\pi_s}$ and $\theta^s_{2,\pi_{s+1}}$ during GMT or FGMT. The cosine similarity tells us how the $(s+1)$-th permutation $\pi_{s+1}$ is consistent with the previous permutation $\pi_s$ on the previous source parameter or trajectory $(\theta^0_1, \cdots, \theta^s_1)$. The results show that Lemma 3.2 actually holds in the early phase of transfer, and also holds in the later phase of FGMT, due to its stability as discussed in Section 3.4.

### E.10 $L^2$ LOSS IN GRADIENT MATCHING

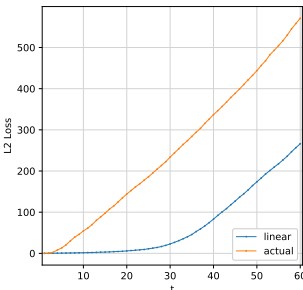

Figure 17: Comparison of $L^2$ losses in FGMT between the linear/actual trajectory on CIFAR-10 with Conv8. The loss is computed by equation (6) for each timestep $t$. The loss for the actual trajectory is accumulated constantly from the beginning, while the loss for the linear one is suppressed in the early phase. The result indicates that, due to its stochasticity, each gradient direction along the actual trajectory is too different from each other to be aligned.

