# OpenReview forum: "Transferring Learning Trajectories of Neural Networks"
_ICLR.cc/2024/Conference — ICLR 2024 poster_

### Official Review · Reviewer_EPpK · 2023-10-29

**Soundness:** 3 good
**Presentation:** 3 good
**Contribution:** 4 excellent
**Rating:** 6
**Confidence:** 3

**Summary:**

The computational cost of training the neural network is high.
To reduce these computational costs, this paper also intends to utilize a well-trained neural network.
Contrast to previous studies, this paper proposes to use the training trajectory of neural networks for model training because it contains a lot of information.
The authors dubbed this problem "learning transfer problem," which is the transfer of trajectories from one initial parameter to another.

To this end, the paper proposes an algorithm that matches the gradient continuously along the trajectory through permutation symmetry.
The authors demonstrate efficiency of the algorithm in two ways to evaluate the validity of the proposed algorithm.

The first is the initialization scenarios that consists of random or pre-trained initialization to assess whether the transferred parameters via “learning transfer” without finetuning can enhance the accuracy.
The other is the fine-tuning scenario to validate whether the transferred parameters can improve the learning speed.

With two scenarios, this paper empirically demonstrates the proposed algorithm can train the model quickly and efficiently.

**Strengths:**

The task, “learning transfer problem” the authors proposed is novel to me.

To address this problem, the authors proposed an algorithm to match the trajectories between source and target, which is seemly convincing. To evaluate the validity of the proposed algorithm, the authors, without any training, conducted an experiment that transfers the calculated parameter to match the trajectory, which performed somewhat successfully.

In addition, as a result of fine-tuning after transferring the parameters, it is revealed that the performance increased very quickly.

**Weaknesses:**

1. There is a lack of motivation albeit the promising results. It is a lack of the evidence whether having the same trajectory between tasks is always good.
2. To experimentally prove that an initialization or architecture affects the similarity more than the dataset, it is necessary to verify it on more datasets.

**Questions:**

1. Does it show the good performance to make the model to have the same trajectory in tasks that are not related to each other?
2. Are these transferred models' ensembles better than scratch ensembles? What is it like from an ECE perspective?
3. In Sec. 3.4, there are an explanation why the transfer of the linear trajectory is more stable and has less variance than the transfer of the actual one. The authors explain that it may be because the actual trajectory contains noisy information. I think the theoretical or empirical evidence is necessary to support the explanation.
4. I think we need to do the work shown in Fig. 5 to select the optimal parameter. Then shouldn't we put this process into the pseudo-code?

---

> ### Author Response · Authors · 2023-11-20
> **Rebuttal (without experiments)**
>
> Thank you for the insightful feedback, and sorry for our delayed response.
>
> Since we are still conducting some additional experiments, we answer the questions that do not require experimental results here. Answers with experimental results will follow after the experiments are completed.
>
> > W1. There is a lack of motivation albeit the promising results. It is a lack of the evidence whether having the same trajectory between tasks is always good.
>
> One of our motivations is to utilizing existing learning trajectories for subsequent trainings. For example, as we demonstrated in Section 4.3, we believe that learning transfer leads to low-cost updating of the foundation model within fine-tuned models in future. Also, we provided Theorem 1 as the theoretical evidence of similarity of learning trajectories in simple 2-MLP case. Aside from whether it's good or not, the similarity of the trajectory actually exists at least in initial phase of training of 2-MLP, so we investigated how we can utilize the similarity. We believe that future work in real application would reveal the effectiveness of utilizing the similarity of trajectories.
>
> > Q1. Does it show the good performance to make the model to have the same trajectory in tasks that are not related to each other?
>
> We sincerely note that, in contrast to transfer learning, we did not consider "tasks that are not related to each other" in our learning transfer problem. More precisely, we consider **a single task and multiple initializations** in learning transfer, in contrast to multiple tasks and a single (pre-trained) initialization in transfer learning. Also, as observed in Section 4.3, we believe that updating the foundation model within fine-tuned models is the most promising application. More precisely, when the source trajectory is trained from poorly generalizing initialization (and thus inherits poor generalization), it acquires better generalization after transferred onto more generalizing initialization. Therefore, **even if the trajectory is same, the difference between source/target initializations may lead to better generalization capability**.
>
> > Q4. I think we need to do the work shown in Fig. 5 to select the optimal parameter. Then shouldn't we put this process into the pseudo-code?
>
> As we noted in Appendix D.3.3, there are only two hyperparameters: batch size B and trajectory length T. The batch size is determined based on the dataset size or number of classes. The trajectory length is determined based on the number of epochs for the source trajectory. However, our method works almost similarly in other hyperparameter values. (We have added ablation results on T in Appendix E.5.) Thus basically they should be simply chosen based on the available computational budget.

---

> > ### Comment · Reviewer_EPpK · 2023-11-22
> > **Thank you for responses**
> >
> > A1. One of our motivations is to utilizing existing learning trajectories for subsequent trainings.
> >
> > I mean, the authors have to explain why utilizing existing trajectories can reduce the cost. I didn't think the explanation in Sec. 4.3 is enough. This is an analysis of the results of the experiment. Because the motivation and the evidence having the same trajectory are lack, I also asked the same question as question 1.
> >
> > A4. There are only two hyperparameters: batch size B and trajectory length T.
> >
> > I think the trajectory length T is just time interval. I wonder how to select optimal timestep without experiments (e.g., it is eight in Fig. 5).

---

> ### Author Response · Authors · 2023-11-21
> **Rebuttal (with additional experiments)**
>
> > W2. To experimentally prove that an initialization or architecture affects the similarity more than the dataset, it is necessary to verify it on more datasets.
>
> Thank you for pointing out it. We have added additional empirical analysis of Assumption (P) in Appendix E.8. If we focus on the results on Cars/CUB for Conv8, ResNet18 and ResNet34, we can see that the results largely depends on model architectures (whether it is a ResNet family or not) rather than datasets themselves. Moreover, from a theoretical viewpoint, Theorem 1 guarantees that datasets or distributions do not matter for 2-MLP. This is another example of architecture-dependence rather than dataset-dependence. Of course, we also observed some dataset-dependences such that CIFAR-10 vs CUB/Cars/ImageNet for Conv8, or CUB/Cars vs ImageNet/iNat2017 for ResNets. However, overall trends still seem to depend on architectures rather than datasets.
>
> > Q2. Are these transferred models' ensembles better than scratch ensembles? What is it like from an ECE perspective?
>
> We have added such evaluations of ensembles in Table 4 in Appendix E.5. In summary, we cannot obtain any informative results from ECE (Empirical Calibration Error) evaluations since ECE metric cannot distinguish between standard-ensemble models and single models. This is possibly because we take an ensemble of only three models. However, we instead introduce another metric "matching probability", which is the probability that a prediction from each member of the ensemble matches the ensembled prediction, i.e., it measures diversity of each member in ensemble. Through this metric, we observed that the diversity in ensemble of transferred models is the same level as that of standard ensemble.
>
> > Q3. In Sec. 3.4, there are an explanation why the transfer of the linear trajectory is more stable and has less variance than the transfer of the actual one. The authors explain that it may be because the actual trajectory contains noisy information. I think the theoretical or empirical evidence is necessary to support the explanation.
>
> We have added more analysis on this hypothesis in Appendix E.10. Although we cannot prove something that noises in the actual trajectory causes the instability, we provide an empirical evidence of the instability. More precisely, we evaluated the L2 losses of gradient matching (Eq. 6 in the main text) for actual/linear transfer. We observed that the loss for actual transfer is constantly accumulated from the beginning, while the loss for the linear one is suppressed in the early phase. This implies that each gradient direction along the actual trajectory is too diverse from each other to be aligned, due to its stochastic nature.

---

> > ### Comment · Reviewer_EPpK · 2023-11-22
> > **Thank you for additional experiments**
> >
> > About W2 and W3,
> >
> > Although the authors have conducted additional explanation, I think it is still lack of evidence.
> >
> > A2. Through this metric, we observed that the diversity in ensemble of transferred models is the same level as that of standard ensemble.
> >
> > Thank you for experiments. I wondered the diversity of ensemble. My question is completely answered.

---

> > > ### Author Response · Authors · 2023-11-22
> > >
> > > > About W2 and W3,
> > > >
> > > > Although the authors have conducted additional explanation, I think it is still lack of evidence.
> > >
> > > It is really helpful if your concerns are clarified in more concrete way. Although we cannot assure that additional experimental results will be provided due to timelimit, but we may improve our paper if there still remains any suspicion or something.
> > >
> > > Anyway, thank you again for engaging in discussion and taking your time for our paper.

---

> > > > ### Comment · Reviewer_EPpK · 2023-11-22
> > > > **About W2 and W3,**
> > > >
> > > > I love the idea of the manuscript. But to argue that this manuscript must be accepted (in other words, to raise the score to eight), I need to be sure that it is applicable in a more diverse architecture and dataset. But as the authors mentioned, we have limited time.

---

> ### Author Response · Authors · 2023-11-22
>
> Thank you for the response to our rebuttal.
>
> > I mean, the authors have to explain why utilizing existing trajectories can reduce the cost. I didn't think the explanation in Sec. 4.3 is enough. This is an analysis of the results of the experiment. Because the motivation and the evidence having the same trajectory are lack, I also asked the same question as question 1.
>
> Although we are still less confident if we understand your concern properly, let us clarify it. There seem to be three aspects in your claim: (1) why utilizing existing trajectories can reduce the cost, and (2) what is the motivation and (3) the evidence having the same trajectory are lack.
>
> On (1), **the reason for possibly reducing the cost is because (i) FGMT efficiently transfers a given trajectory with lightweight linear optimization and a few iterations of gradient computation (see Appendix D.3.4) and (ii) the subsequent training is accelerated as shown in Section 4.2**.  Of course, note that we have never claimed that our method provides actual speed up, which remains for future work.
>
> On (2), we do not understand what "the motivation having the same trajectory" exactly means, and so instead we explained the motivation utilizing the existing trajectory in our rebuttal. Also we discussed about it in Introduction.
>
> On (3), we provided Theorem 1 for theoretical evidence and Figure 15 for empirical evidence, while we consider Assumption (P) should be somewhat relaxed for modern architectures beyond MLPs in future work, as discussed in Section 3.2.
>
> > I think the trajectory length T is just time interval. I wonder how to select optimal timestep without experiments (e.g., it is eight in Fig. 5).
>
> For the first sentence, yes. For the second sentence, we need to clarify that the x-axis of Figure 5 is not T (the length of trajectory), but just timestep t. So **we can fix T before transfer (as explained in our rebuttal), and run the FGMT algorithm until the validation accuracy starts to drop**. In other words, we can choose an (nearly) optimal timestep just like well-known early stopping. This is also explicitly explained in Section 4.1.

---

> ### Author Response · Authors · 2023-11-23
>
> Thank you for the constructive feedback. We respectfully notice you that the current manuscript also contains experimental results with additional real-world datasets (SUN-397, iNaturalist) and an additional larger network (ResNet-34), through our rebuttal to other reviewers. More additional results have been summarized in the general response. We appreciate it if you reassess your evaluation based on these consistent results and on what we claimed as our contributions. Thank you again for your time for reviewing.

---

### Official Review · Reviewer_rULw · 2023-10-29

**Soundness:** 3 good
**Presentation:** 3 good
**Contribution:** 3 good
**Rating:** 8
**Confidence:** 3

**Summary:**

### Problem Statement

The paper introduces a novel problem called the "learning transfer problem", aimed at reducing training costs for seemingly duplicated training runs on the same dataset by transferring a learning trajectory from one initial parameter to another without actual training.

### Main Contribution

The paper's contributions include: (1) formulating the new problem of learning transfer with theoretical backing, (2) deriving the first algorithm to solve it, (3) empirically demonstrating that transferred parameters can accelerate convergence in subsequent training, and (4) investigating the benefits and inheritances from target initializations. Through these contributions, the paper presents a promising avenue to significantly reduce the computational cost and time required to train DNNs, especially in scenarios involving model ensembles or fine-tuning pre-trained models.

### Methodology

The authors approximate the solution to the learning transfer problem by matching gradients successively along the trajectory via a permutation symmetry technique. The updates along the "source trajectory" are applied to update a target network with a different random initialization after being permutated such that the gradients of the two networks at the same "time step" are best matched under the permutation, resulting in the Gradient Matching along Trajectory (GMT) algorithm. To further optimize the space and time complexity of the algorithm, the authors propose to use linear interpolation of the initial and final parameters of the source network in place of the acutal training trajectory, and to re-use the mini-batch gradients evaluated along the trajectories to search for the best-matching permutations. The optimized verison of the algorithm is named "Fast Gradient Matching along Trajectory" (FGMT). The best permutations for parameter alignment at each time step are solved with a coordinate descent algorithm, iteratively optimizing for the permutation in each layer of the network by solving a linear assignment problem.

### Experiments

The learning transfer methods are evaluated on standard vision datasets with various architectures including CNN and MLP. Both random initializations and pre-trained initializations are used to evaluate and demonstrate the effect of GMT and FGMT in terms of 1) the performance after transfer; 2) the fine-tuning efficiency and performance after transfer.

Empirical evaluations reveal that the transferred parameters achieve non-trivial accuracy before any direct training and can be trained significantly faster than training from scratch, while inheriting the properties (e.g. generalization ability) from the parameter initialization.

**Strengths:**

### Originality and significance

The proposed task of learning transfer problem is novel and very interesting, with potentially wide applications, as the foundation-model paradigm prevails in many AI / DL fields. The proposed method is to progressively merge the target network with the source network using Git Re-basin, which is straightforward and efficient.

### Quality

Theoretical analysis is performed to justify the adopted method, in addition to a series insightful experiments. The experimental details are in general well documented. However, I find the experiments are not enough to support some of the claims, and will expand on this in the Weakness section.

### Writing

The writing is overall good, despite minor grammar problems. I find the mathematics in the paper is clear with consistent and self-explanatory notations. The problem is well motivated and formulated, and the main thesis is well conveyed.

**Weaknesses:**

I am open to change my score if the authors can address the following concerns:

### Lack of experiments more closely demonstrating the actual usage of the proposed method

1. One potential usage of the proposed method suggested by the authors is to transfer the update of a foundation model to its fine-tuned versions. However, all experiments are limited to network architectures of relatively smaller scale, and to the cases where fine-tuning task shares exactly the same number of classes as the pre-training task, which differs from the realistic use-case of foundation models, which are of typically larger scale, and are used for various down-stream tasks with task-specific heads.

2. The authors claim that method can accelerate the training of an ensemble of neural networks. Although the computational cost of the proposed method is briefly described in the appendix, there is no figure or table systematically comparing the cost with traditional training / fine-tuning approaches. More crucially, no experiment compares the performance of the ensemble obtained with GMT / FGMT and traditional approaches. One concern is that, transfering one (or a limited number of) source trajectory, the diversity of the resulted target networks is limited (this is partially endorsed by the landscape visualization in Figure 7), which could hurt the performance of the ensemble.

### Lack of experiments verifying the Assumption (P), Theorem 3.1, and Lemma 3.2

Although the theoretical analysis abounds, it would be much more convincing to show these statements hold in real experiments.

### Lack of experiments and discussion on the choice of $T$

It is not clear how $T$, the numebr of time steps, or rather, the number of samples taken from the source trajectory, influences the performance of the transfer result. It seems that $T$ does not really matter in the Naïve baseline and Oracle baseline, where the parameter-aligning permutation $\pi$ remains the same across time steps, which 1) would be good to be verified by the authors in the main text and 2) makes it interesting to explore the value of $T$ that GMT / FGMT requires to have good performance, because the computational cost is proportional to $T$.

### The "Generalization ability" part in section 4.3 is not clear

In the "Generalization ability" part in section 4.3, the experiments and figures should be further clarified. From Figure 7, I assume that the task is to fit the CUB or Cars dataset (for the previously chosen 10-class subset), and the two "Standard" curves are for the target trajectories, which are transferred through FGMT to initialization pretrained with ImageNet-10%, leading to the green curves, but the explanations are really not clear. However, the main text explicitly states that the target initialization $\theta_2^0$ is "pre-trained on full ImageNet", which implies only *1* (instead of *2*) possible combination for FGMT for each fine-tuning dataset: starting from ImageNet-10%-pretrained, transferring the finetuning trajectory which starts from the ImageNet-Full-pretrained initialization). Another example is, I am not sure what validation the authors refer to when they mention that the ImageNet-Full pretrained initialization has "validation accuracy $\approx$ 72%" and the ImageNet-10% one has "validation accuracy $\approx$ 50%".


### Minor
- In the second last line of the second paragraph in Introduction: strength -> strengthen
- On Page 5, right below Lemma 3.2: please make it explicit in which Appendix the proof is
- On Page 6, in the "Linear trajectory" section, "such a linearly interpolated trajectory *satisfies* ..."
- On Page 9, in the "Generalization ability" section, "generalizes poorly *than* ..." -> "generalizes poorly *compared to* ..."

**Questions:**

1. Data augmentation is widely used for the training of DNNs, which essentially (often randomly) modifies the dataset. Does this violate the assumption that the dataset is the same in the problem statement?

2. Even gradient-based, many optimization methods for DL can have updates very different from SGD (for example, Adam). Considering that the parameter alignment is done with gradient matching in GMT and FGMT, does the choice of optimization method changes the effect of GMT / FGMT? Ablation study regarding this could be worthwhile to add.

3. Why does the performance deteriorate in Figure 5d-5f? It is simply acknowledged without analysis and discussion.

4. In Figure 7, to better understand the effect of FT and FGMT, where would the FGMT-only parameters be in the landscape? What about permuted-source + FT?

---

> ### Author Response · Authors · 2023-11-20
> **Rebuttal (without experiments)**
>
> Thank you for the insightful and detailed feedback, and sorry for our delayed response.
>
> Since we are still conducting some additional experiments, we answer the questions that do not require experimental results here. Answers with experimental results will follow after the experiments are completed.
>
> > 1. One potential usage of the proposed method suggested by the authors is to transfer the update of a foundation model to its fine-tuned versions. However, all experiments are limited to network architectures of relatively smaller scale,
>
> Due to our limited resources, we could not add experiments for foundation models. However, our theoretical result (Theorem 1) provides an evidence that learning transfer works more effectively in the case of networks with more neurons. Intuitively, more neurons give us more degree of freedom to match them by some permutation. Also, we emphasize that our paper should be considerred as the first step to the proposed new problem, where we demonstrated that learning transfer is indeed possible for MLPs/CNNs/ResNets and promising for updating pre-trained models (Section 4.3). Extending our approach to more complex architectures such as Transformers or Diffusion Models needs more work (for e.g., how to exploit the symmetry of self-attention, how to deal with the inverse diffusion process, etc), so it remains for future work.
>
> > and to the cases where fine-tuning task shares exactly the same number of classes as the pre-training task, which differs from the realistic use-case of foundation models, which are of typically larger scale, and are used for various down-stream tasks with task-specific heads.
>
> Sorry for the confusion. We notice that we have forgotten to remove the statement "a 10-classes subset of CIFAR-100" in Section 4.1 from pre-submitted version of our paper, since we are using CIFAR-100 with 100 classes in the submitted version. Thus, we sincerely note that, in all experiments of fine-tuning scenario, fine-tuning task **does not** share the same number of classes as the pre-training task. Indeed, before transferring learning trajectories, we replace the final layer of the given (target) pre-trained initialization by a task-specific head, as well as the standard way. Sorry again for confusing you by the misprint.
>
> > In the "Generalization ability" part in section 4.3, the experiments and figures should be further clarified.
>
> Sorry for the confusion. We have added more explicit explanations in Section 4.3 in the latest revision. Briefly, we investigated the fine-tuning scenario (i.e., transferring a learning trajectory starting from pre-trained models) where the source initialization $\theta_1^0$ has worse generalization ability (pre-trained on 10% subset of ImageNet) than target initialization $\theta_2^0$ (pre-trained on full ImageNet). The learning trajectory starting from the former initialization should have also worse generalization. However, just by transferring the trajectory onto the latter initialization, the transferred trajectory obtains better generalization inherited from the latter initialization (even though it has never been trained on the latter initialization!).
>
> > Another example is, I am not sure ... "validation accuracy \approx 72%" and the ImageNet-10% one has "validation accuracy \approx 50%".
>
> These are validation accuracies of the pre-trained models on ImageNet. We have added this explanation in the main text.
>
> > Q1. Data augmentation is widely used for the training of DNNs, which essentially (often randomly) modifies the dataset. Does this violate the assumption that the dataset is the same in the problem statement?
>
> We treat data augmentation as just noise in gradients as well as mini-batch order. Basically it draws samples from the same distribution, so it should not matter.
>
> > Q3. Why does the performance deteriorate in Figure 5d-5f? It is simply acknowledged without analysis and discussion.
>
> In the case of pre-trained initializations, we can see that Assumption (P) starts to fail in very early iterations in Figure 3. This causes the performance degradation in Figure 5d-5f, which are transfer between pre-trained initializations. Addressing this challenge also remains as future work.
>
> > Q4. In Figure 7, to better understand the effect of FT and FGMT, where would the FGMT-only parameters be in the landscape? What about permuted-source + FT?
>
> For the first question, there are two reasons: (1) since the accuracy of FGMT-only is less than source or target parameters, plotting FGMT-only parameter gives us no insightful information about their relationships in loss landscape, (2) in Section 4.3, we aimed to investigate the nature of FGMT+FT parameter since it is more important than FGMT-only parameter in practice. For the second question, since permuting a NN does not change its output or accuracy, we do not need to train the permuted-source further. If we did so, it would not change the loss landscape property of the permuted-source parameter.

---

> > ### Comment · Reviewer_rULw · 2023-11-21
> >
> > Thanks for the response. For Figure 7, I believe it would be beneficial to more explicitly clarify. My current understanding is, for example, in the CIFAR10 -> CIFAR100 case, "Target" means a network pretrained with CIFAR100, and finetuned for the CIFAR10 task, while the "Source" means a network trained for the CIFAR10 task, but not starting from a pretrained initialization (is it random initialization then?), FGMT means the result of transferring the training trajectory of "Source" to the CIFAR100-pretrained initialization, and the "validation accuracy" in this case refers to the accuracy evaluated on CIFAR10 validation set. This in my humble opinion is by no means obvious, and some more notations or examples clarifying what each term refers to could really help. Still, I'm not convinced that FGMT-only parameters cannot provide insightful information wrt where in the landscape that the trajectory transferring can take the parameters to. I would even argue that including the initialization points (both pretrained and non-pretrained) in the landscape plot would further clarify the intuition about how initialization, finetuning / training, and trajectory transfer influence where in the parameter space the network end up with.

---

> > > ### Author Response · Authors · 2023-11-21
> > >
> > > Thank you for the quick response, and sorry for the confusion again. We have clarified the descriptions for Source and Target in the updated revision. In summary, "Target" refers to the parameter fine-tuned from a target initialization as you said, but "Source (Permuted)" refers to the end point of the source trajectory (which is **the parameter fine-tuned from a source initialization**) permuted by Git Re-basin. Therefore, both of them are fine-tuned on CIFAR100/CUB/Cars. Also, while Target and Source have almost same (optimal) accuracy on these datasets, the FGMT-only parameter is not optimal and thus should be fine-tuned (i.e. FGMT+FT) to properly compare with Target and Source in the landscape.

---

> ### Author Response · Authors · 2023-11-21
> **Rebuttal (with additional experiments)**
>
> > 2. The authors claim that method can accelerate the training of an ensemble of neural networks.
>
> We sincerely note that we have made no such a claim on ensemble. Sorry for the confusion. At least, we did not intend to accelerate training of ensemble by the proposed method as is. Indeed, the results of ensemble is not always as good as the standard ones as provided in Appendix E.5. However, we believe that future method may achieve such acceleration as discussed in Introduction due to the lightweight nature of learning transfer.
>
> > Although the computational cost of the proposed method is briefly described in the appendix, there is no figure or table systematically comparing the cost with traditional training / fine-tuning approaches.
>
> We have added a table on wall-clock time for transfer in Appendix E.6. However, we would like to emphasize that the results are highly unreliable because computing nodes are shared with other research
> groups and has non-negligible I/O delay due to its distributed storage. Also our implementation is not so optimized. This is because our main purpose in this paper is to demonstrate the possibility and potential applicability of learning transfer, as we claimed in the last lines of Introduction. Actual speed-up for ensembles or foundation-updates is promising challenges for future research.
>
> > One concern is that, transfering one (or a limited number of) source trajectory, the diversity of the resulted target networks is limited (this is partially endorsed by the landscape visualization in Figure 7), which could hurt the performance of the ensemble.
>
> In Table 4 of Appendix E.5, we have added results on diversity of transferred models in ensemble, compared to standard ensemble. Although we evaluated calibration error as Reviewer EPpK suggested, it fails to evaluate the diversity because it cannot distinguish between single models and standard ensembles (possibly because we take an ensemble of only three models). Instead, we introduce another metric "matching probability", which is the probability that a prediction from each member of the ensemble matches the ensembled prediction, i.e., it measures diversity of each member in ensemble. Through this metric, we observed that the diversity in ensemble of transferred models is the same level as that of standard ensemble.
>
> > Lack of experiments verifying the Assumption (P), Theorem 3.1, and Lemma 3.2
>
> For Assumption (P) and Theorem 3.1, we have added additional results in Appendix E.8 with more datasets and architectures. Overall trends are consistent with Figure 3 in the main text, which also provides empirical analysis for Assumption (P) and Theorem 3.1.
>
> For Lemma 3.2, we have added empirical analysis in Appendix E.9. We evaluated the cosine similarity between $\theta_{2,\pi_s}^s$ and $\theta_{2,\pi_{s+1}}^s$ during GMT or FGMT, which are expected to be similar to each other by Lemma 3.2. The results show that Lemma 3.2 (with $t=s$ in the statement) actually holds in the early phase of transfer, and also holds in the later phase of FGMT, due to its stability as discussed in Section 3.4.
>
> > It seems that T does not really matter in the Naïve baseline and Oracle baseline, ..., which 1) would be good to be verified by the authors in the main text
>
> Thanks for pointing out it. We have added such explanation in Baselines paragraph (Section 4).
>
> > 2) makes it interesting to explore the value of T that GMT / FGMT requires to have good performance, because the computational cost is proportional to T.
>
> Thanks for the suggestion. We have added some experiments with varying T in Appendix E.7. The results indicate that too small T (such as T=5) makes the transfer a bit worse, but it stops improving validation accuracy beyond some T (such as T=15 or 20 in our experiments). In summary, the choice of T slightly influences the results of our algorithm if T is enough size.
>
> > Q2. Even gradient-based, many optimization methods for DL can have updates very different from SGD (for example, Adam). Considering that the parameter alignment is done with gradient matching in GMT and FGMT, does the choice of optimization method changes the effect of GMT / FGMT? Ablation study regarding this could be worthwhile to add.
>
> Thank you for the interesting suggestion. We have added some results with Adam in Appendix E.4. The results show that (1) during transfer it fails to achieve high accuracy in contrast to SGD cases, (2) but it still accelerates a bit the subsequent training. We hypothesize that this is related to the phenomenon that MLI (Monotonic Linear Interpolation) fails in Adam cases as previous work [1] have observed, which indicates the trajectory of Adam is more complex than that of SGD. Addressing this difficulty with Adam is also an interesting direction in future work.
>
> [1] Lucas et al. "On monotonic linear interpolation of neural network parameters" (ICML'21)

---

> ### Comment · Reviewer_rULw · 2023-11-23
>
> Thanks for the comprehensive response and a wide range of experiments added. They clarified most of my questions.
>
> A minor problem with respect to Figure 7 again: In the caption of Figure 8, the notation "X -> Y" means the training trajectory on task X is transferred to initialization pretrained on task Y, while in the author's clarifying response, in the "CIFAR10 -> CIFAR100" case, the "Source" refers to the end point of the source trajectory fine-tuned on CIFAR100. These notations seem inconsistent and rather confusing.
>
> Overall it is an interesting work with supporting experiments, while the applicability and practical value of the proposed method are still limited or not fully proven, and I would keep my rating of 6 unchanged.

---

> ### Author Response · Authors · 2023-11-23
>
> Thank you for the response.
>
> > A minor problem with respect to Figure 7 again: In the caption of Figure 8, the notation "X -> Y" means the training trajectory on task X is transferred to initialization pretrained on task Y, while in the author's clarifying response, in the "CIFAR10 -> CIFAR100" case, the "Source" refers to the end point of the source trajectory fine-tuned on CIFAR100. These notations seem inconsistent and rather confusing.
>
> Actually, this has been already fixed in the current manuscript with new notation "X => Y" (but silently) based on your initial review comment. Thank you again for pointing out the ambiguity around Figure 8!
>
> > the applicability and practical value of the proposed method are still limited or not fully proven
>
> To this point, we respectfully note that we have never claimed the applicability of *our proposed method* itself as our contribution (as you also summarized in your review). We consider our work as a starting point of the new problem, and thus the applicability of the proposed method is still limited as you pointed out. Rather, our purpose of this paper is to demonstrate the learning transfer phenomenon and its potential applicability, i.e., not applicability of the proposed method. We also greatly appreciate it if you reassess your evaluation based on such aspects of our work, and also based on our general response. Thank you again for valuable feedback through this discussion.

---

### Official Review · Reviewer_gCAh · 2023-11-01

**Soundness:** 2 fair
**Presentation:** 3 good
**Contribution:** 3 good
**Rating:** 6
**Confidence:** 3

**Summary:**

This paper proposes a novel algorithm for transferring a learning trajectory from one initial parameter to another, which can significantly reduce the computational cost of training deep neural networks. The algorithm formulates the learning transfer problem as a non-linear optimization problem for the policy function and matches gradients successively along the trajectory via permutation symmetry to approximately solve it. The empirical results show that the transferred parameters achieve non-trivial accuracy before any direct training and can be trained significantly faster than training from scratch. However, the algorithm's limitations include the assumption that the source and target tasks are related, and the lack of a detailed analysis of the computational cost of the algorithm.

**Strengths:**

1. The proposed algorithm is a novel approach to the problem of transferring a learning trajectory from one initial parameter to another. The idea is interesting.
2. The algorithm is theoretically grounded and can be solved efficiently with only several tens of gradient computations and lightweight linear optimization.
3. The empirical results show that the transferred parameters achieve non-trivial accuracy before any direct training and can be trained significantly faster than training from scratch.

**Weaknesses:**

1. The empirical evaluation of the algorithm is conducted on a limited set of benchmark datasets, and it is unclear how well the algorithm would perform on other types of datasets or in real-world scenarios.
2. The paper assumes that the source and target tasks are related, and it is unclear how well the algorithm would perform when the tasks are not such related.
3. The paper does not provide a detailed analysis of the computational cost of the algorithm, which may be a concern for large-scale neural networks.
4.

**Questions:**

1. How sensitive is the algorithm's performance to the assumption that the source and target tasks are related, and how well does it perform when the tasks are unrelated?
2. How does the proposed algorithm compare to other methods for transferring learning trajectories, such as fine-tuning or transfer learning?
3. How well does the algorithm perform on datasets that are not included in the empirical evaluation, and how does its performance compare to other methods on these datasets?
4. Can the algorithm be extended to handle more complex neural network architectures?
5. How does the computational cost of the algorithm compare to other methods for transferring learning trajectories, and how does it scale with the size of the neural network and the number of training epochs?

---

> ### Author Response · Authors · 2023-11-20
> **Rebuttal (without experiments)**
>
> Thank you for the fruitful feedbacks, and sorry for our delayed response.
>
> Since we are still conducting some additional experiments, we answer the questions that do not require experimental results here. Answers with experimental results will follow after the experiments are completed.
>
> > W2. The paper assumes that the source and target tasks are related, and it is unclear how well the algorithm would perform when the tasks are not such related.
>
> We sincerely note that, in contrast to transfer learning, there are no "source and target tasks" in our learning transfer problem. More precisely, we consider **a single task and multiple initializations** in learning transfer, in contrast to multiple tasks and a single (pre-trained) initialization in transfer learning. The difference of the problem itself is one of our contributions, as other reviewers acknowledge it. Nevertheless, it would be an interesting future direction to investigate how to extend our problem with **multiple tasks** and multiple initializations, which may integrate transfer learning and learning transfer.
>
> > Q1    How sensitive is the algorithm's performance to the assumption that the source and target tasks are related, and how well does it perform when the tasks are unrelated?
>
> See answer to W2.
>
> > Q2    How does the proposed algorithm compare to other methods for transferring learning trajectories, such as fine-tuning or transfer learning?
>
> See answer to W2. Fine-tuning and transfer learning are not designed for solving our problem, and it is also not obvious how to exploit them in our problem.
>
> > Q4    Can the algorithm be extended to handle more complex neural network architectures?
>
> The algorithm only exploits the permutation symmetry of intermediate neurons in feed-forward MLPs, so in principle our method can be applied to many network architectures that can be seen as a restricted version of MLPs like CNNs. However, it remains as an important future work how to deal with network architectures beyond feed-forward MLPs, such as Transformers, which involve additional symmetery from their attention structure.

---

> ### Author Response · Authors · 2023-11-21
> **Rebuttal (with experimental results)**
>
> Here, we answer to weaknesses/questions with additional experimental results.
>
> > W1.  The empirical evaluation of the algorithm is conducted on a limited set of benchmark datasets, and it is unclear how well the algorithm would perform on other types of datasets or in real-world scenarios.
>
> We have added additional experiments on more large-scale and complex datasets (SUN397 and iNaturalist2017) than Cars and CUB in Appendix E2. The results show that (1) transfer-only achieves less accuracy compared to Cars/CUB cases but (2) it still accelerates the subsequent training after transfer a bit, which indicates the future possibility of improving our method for complex datasets. Also, we would like to emphasize that Theorem 1 does not restrict the type of datasets or distributions at least in 2-MLP case. Thus we believe that learning transfer will be effective for real-world datasets if future research continues to improve methods.
>
> > W3.  The paper does not provide a detailed analysis of the computational cost of the algorithm, which may be a concern for large-scale neural networks.
>
> In Appendix D.3.4, we briefly discussed the computational cost of our algorithms. Also we have added wall-clock time for learning transfer in Appendix E.6. The results also include the comparison between ResNet18 and ResNet34, from which we can see how computational cost grows for large networks in real experiments. For more theoretical aspects, see also answer to Q5. However, it is worth noting that **our main contributions are (1) problem formulation, (2) derivation of algorithm (3) empirical demonstration of learning transfer (4) analysis of transferred solutions**, and thus not an actual speed-up in real world applications, as we explicitly claimed in Introduction.
>
> > Q3    How well does the algorithm perform on datasets that are not included in the empirical evaluation, and how does its performance compare to other methods on these datasets?
>
> See answer to W1.
>
> > Q5    How does the computational cost of the algorithm compare to other methods for transferring learning trajectories,
>
> See answer to W2 for "other methods for transferring learning trajectories".
>
> > and how does it scale with the size of the neural network and the number of training epochs?
>
> The algorithm mainly constitutes of gradient computation (GC) and gradient matching (GM). Roughly speaking, for the neural network with $N$ neurons, GC requires $O(N^2)$ computation steps as well-known with backpropagration, and GM requires $O(N^3)$ steps with Hungarian algorithm. Also the algorithm does not depend on the number of epochs, but rather on the length of trajectory T and batch size B, which should be chosen by computational budgets. See also answer to W3.

---

### Author Response · Authors · 2023-11-23
**Summary of our revisions**

Dear Reviewers, Area Chairs and Senior Area Chairs,

We would like to thank you for taking your time to review our paper, and for your valuable feedback. After rebuttal, the current manuscript contains several additional results, which support or supplement our claims. Here we briefly summarize them:

- Appendix E.2 & E.3: Experiments with **additional real-world datasets (SUN-397, iNaturalist)** and **an additional larger architecture (ResNet-34)**.
- Appendix E.4: Experiments with **Adam optimizer**.
- Appendix E.5: **Diversity analysis in ensemble of transferred models**.
- Appendix E.6: **Benchmarks of wall-clock time in learning transfer**.
- Appendix E.7: **Sensitivity analysis to the hyperparameter T**.
- Appendix E.8: **Additional empirical verification** for Assumption (P).
- Appendix E.9: **Empirical validation for Lemma 3.2**.
- Appendix E.10: Comparison of optimization of Eq.(6) between actual vs linear trajectories.

Finally, we would like to emphasize again that **the main contributions are as claimed in Introduction: (1) problem formulation with theoretical backing, (2) derivation of algorithm, (3) empirical demonstration of learning transfer, (4) analysis of transferred solutions in terms of generalization and loss landscape**, and nothing more. **These claims are validated by theoretical or empirical analysis including the above additional results.** Nevertheless, the proposed method should be further improved for actual speed-up in real-world applications with modern architectures like Transformer in future work. However, we consider that this does not weaken the claimed contributions.

Thanks again to everyone involved in the review process.

---

### Meta-Review · Area_Chair_bgaW · 2023-12-10

**Metareview:**

The paper introduces and tackles for the first time an ambitious "learning transfer" problem, where the objective is to transfer a learning trajectory from one parameter from a well trained network to another from an untrained one. Such a problem if solved can significantly reduce the computational cost of training deep neural networks. The algorithm proposed formulates the learning transfer problem as a non-linear optimization problem for the policy function and matches the gradient continuously along the trajectory through permutation symmetry. The empirical results demonstrated the proposed algorithm can train the model quickly and efficiently.

**Justification For Why Not Higher Score:**

While proposing an interesting and novel problem and an initial attempt of a solution, the methodology is somewhat derivative to the existing work of Git re-basin. The experiments conducted are also of limited architecture and datasets.

**Justification For Why Not Lower Score:**

The paper stated a novel problem as well as a proposed solution. Setting the ambitious goal of learning transfer would potentially encourage more researchers to tackle this problem, and greatly improve the training efficiency of the current technology.

The paper contains both theoretical and empirical analyses, shows promising initial results, with plenty of ablation and supplementary studies.

---

### Decision · Program_Chairs · 2024-01-16

Accept (poster)